# A Parameter-Free and Near-Optimal Zeroth-Order Algorithm for Stochastic Convex Optimization

**Kunjie Ren** [1]   **Luo Luo** [1] [2]

## Abstract

This paper studies zeroth-order optimization for stochastic convex minimization problems. We propose a parameter-free stochastic zeroth-order method (POEM), which introduces a stepsize scheme based on the distance over finite difference and an adaptive smoothing parameter. Our theoretical analysis shows that POEM achieves near-optimal stochastic zeroth-order oracle complexity. Furthermore, numerical experiments demonstrate that POEM outperforms existing zeroth-order methods in practice.

## 1. Introduction

This paper studies the stochastic optimization problem

$$\min_{\mathbf{x} \in \mathcal{X}} f(\mathbf{x}) \triangleq \mathbb{E}_{\boldsymbol{\xi} \sim \Xi}[F(\mathbf{x}; \boldsymbol{\xi})] \tag{1}$$

where the domain $\mathcal{X} \subseteq \mathbb{R}^d$ is a compact convex set, the random variable $\boldsymbol{\xi}$ follows a distribution $\Xi$, and the stochastic component function $F(\mathbf{x}; \boldsymbol{\xi})$ is convex and Lipschitz continuous in $\mathbf{x}$ over $\mathcal{X}$ for any given $\boldsymbol{\xi}$. We focus on stochastic zeroth-order optimization for solving Problem (1), where the algorithm can only query stochastic function values. This setting is particularly relevant when accessing (stochastic) first-order information is expensive or infeasible. Such scenarios arise in various applications, including bandit optimization (Flaxman et al., 2004; Agarwal et al., 2010; Shamir, 2017), adversarial training (Goodfellow et al., 2014; Shaham et al., 2018), reinforcement learning (Balasubramanian & Ghadimi, 2018; Mania et al., 2018), and other black-box models (Liu et al., 2016; Ilyas et al., 2018).

Finite difference methods are widely used in zeroth-order optimization, where they estimate the first-order information of the objective function via random directions (Kiefer & Wolfowitz, 1952; Ghadimi & Lan, 2013; Duchi et al., 2015; Nesterov & Spokoiny, 2017; Nazari et al., 2020; Gasnikov et al., 2022; Lin et al., 2022; Rando et al., 2024; Chen et al., 2023; Kornowski & Shamir, 2024). For stochastic convex problems with Lipschitz continuous components, Nesterov & Spokoiny (2017) first developed random search methods that achieve suboptimal convergence rates. Subsequently, Duchi et al. (2015) proposed a stochastic algorithm that constructs finite differences using two random sequences, improving the dependence on problem dimension compared to Nesterov & Spokoiny (2017). They also established a lower bound, demonstrating that their algorithm achieves near-optimal stochastic zeroth-order oracle (SZO) complexity. Building on this, Shamir (2017) introduced an algorithm using a single random sequence drawn from the uniform distribution over the unit ball, which is optimal and easy to implement. Finite difference methods have also been broadly applied to smooth optimization problems (Ghadimi & Lan, 2013; Nesterov & Spokoiny, 2017; Duchi et al., 2012b; Balasubramanian & Ghadimi, 2018), as well as to nonsmooth and nonconvex settings (Lin et al., 2022; Chen et al., 2023; Kornowski & Shamir, 2024). Despite these advances, existing zeroth-order optimization methods (Kiefer & Wolfowitz, 1952; Ghadimi & Lan, 2013; Duchi et al., 2015; Nesterov & Spokoiny, 2017; Nazari et al., 2020; Gasnikov et al., 2022; Lin et al., 2022; Rando et al., 2024) face several limitations. A key challenge is their high sensitivity to parameter settings. Achieving optimal convergence rates typically requires carefully tuned step sizes that depend on prior knowledge of problem properties (such as the Lipschitz constant) and the iteration budget. In addition, the smoothing parameter used in the finite difference often depends on the target accuracy or decays rapidly, which can lead to numerical instability in practice.

We desire to develop adaptive stochastic optimization methods that remove the need of parameter tuning. Most existing work focus on first-order methods. For example, Rolinek & Martius (2018); Vaswani et al. (2019); Paquette & Scheinberg (2020) introduced line search techniques for stochastic optimization. Tan et al. (2016); Berrada et al. (2020); Loizou et al. (2021); Wang et al. (2023) extended the Barzilai–Borwein (BB) step size (Barzilai & Borwein, 1988) and the

---

[1]School of Data Science, Fudan University, Shanghai, China [2]Shanghai Key Laboratory for Contemporary Applied Mathematics, Shanghai, China. Correspondence to: Luo Luo <luoluo@fudan.edu.cn>.

*Proceedings of the 42$^{nd}$ International Conference on Machine Learning*, Vancouver, Canada. PMLR 267, 2025. Copyright 2025 by the author(s).

Polyak step size (Polyak, 1987) to stochastic settings. For training deep neural networks, adaptive algorithms such as AdaGrad (Duchi et al., 2011), Adam (Kingma & Ba, 2014), and their variants (Tieleman, 2012; Zeiler, 2012; Shazeer & Stern, 2018; Wang et al., 2024; Zhang et al., 2025) exploited the specific problem structure and have achieved success in many applications. However, these methods still rely on appropriately chosen initialization parameters, which can significantly influence convergence behavior, both in theory and in practice.

Ideally, we aim to design a parameter-free optimization method that achieves near-optimal convergence rates while requiring minimal knowledge of problem-specific properties (Streeter & McMahan, 2012; Defazio & Mishchenko, 2023; Lan et al., 2023; Li & Lan, 2023). Several parameter-free methods for stochastic convex optimization have been developed using online learning techniques (Luo & Schapire, 2015; Orabona & Pál, 2016; Cutkosky & Orabona, 2018; Bhaskara et al., 2020; Mhammedi & Koolen, 2020; Jacobsen & Cutkosky, 2022), though their implementations are often quite complex. In practice, Orabona & Tommasi (2017); Chen et al. (2022) applied coin-betting techniques within the classical stochastic gradient descent (SGD) framework, achieving strong empirical performance in training neural networks. Later, Carmon & Hinder (2022) showed that using a bisection step to adaptively determine the step size in SGD yields a parameter-free algorithm with near-optimal convergence rates. Building on this work, Ivgi et al. (2023a) proposed a parameter-free step size schedule called Distance over Gradients (DoG), which adjusts step sizes based on the distance from the initial point and the norm of stochastic gradients (You et al., 2017; Shazeer & Stern, 2018; Bernstein et al., 2020). DoG achieves near-optimal convergence rates and performs well in practice. However, all existing parameter-free methods are designed for first-order optimization. Extending these methods to the zeroth-order setting presents additional challenges, particularly the need to eliminate tuning for both the step size and the smoothing parameter, as well as to carefully control the dependence on the problem dimension in the convergence rates.

In this paper, we propose a parameter-free stochastic zeroth-order method (POEM), which introduces a stepsize scheme based on the distance over finite difference and an adaptive smoothing parameter. For the stochastic convex optimization (1), we show that the initialization affects the convergence rates only by a logarithmic factor. We establish high-probability convergence guarantees, demonstrating that POEM achieves near-optimal SZO complexity. A comparison of POEM with related methods is presented in Table 1. We also study the problems with unbounded domains and show that an ideal parameter-free algorithm is impossible in such settings. Finally, we conduct numerical experiments to validate the practical effectiveness of POEM.

## 2. Preliminaries

In this section, we formalize the problem setting and introduce the smoothing technique in zeroth-order optimization.

### 2.1. Notation and Assumptions

Throughout this paper, we use $\|\cdot\|$ to denote the Euclidean norm. The unit ball is defined as $\mathbb{B}^d \triangleq \{\mathbf{u} \in \mathbb{R}^d : \|\mathbf{u}\| \leq 1\}$ and the unit sphere as $\mathbb{S}^{d-1} \triangleq \{\mathbf{v} \in \mathbb{R}^d : \|\mathbf{v}\| = 1\}$. We denote by $\mathbb{U}(\mathbb{B}^d)$ and $\mathbb{U}(\mathbb{S}^{d-1})$ the uniform distributions on the unit ball and the unit sphere, respectively. Additionally, we use the notation $\tilde{\mathcal{O}}(\cdot)$ to suppress logarithmic factors.

We make the following assumptions for Problem (1).

**Assumption 2.1.** The domain $\mathcal{X} \subseteq \mathbb{R}^d$ is compact and convex. Furthermore, we denote the diameter of $\mathcal{X}$ by

$$D_{\mathcal{X}} \triangleq \max_{\mathbf{x}, \mathbf{y} \in \mathcal{X}} \|\mathbf{x} - \mathbf{y}\| < \infty.$$

Next, we define the Euclidean projection onto the domain.

**Definition 2.2.** For any point $\mathbf{x} \in \mathbb{R}^d$, the Euclidean projection onto the compact convex set $\mathcal{X} \in \mathbb{R}^d$ is given by

$$\Pi_{\mathcal{X}}(\mathbf{x}) \triangleq \arg\min_{\mathbf{y} \in \mathcal{X}} \|\mathbf{x} - \mathbf{y}\|.$$

Under Assumption 2.1, the objective function attains its minimum over the compact set $\mathcal{X}$. Hence, we define the optimal solution to Problem (1) as follows.

**Definition 2.3.** Let $\mathbf{x}_{\star} \in \mathcal{X}$ be an optimal solution to Problem (1) such that $f(\mathbf{x}_{\star}) = \min_{\mathbf{x} \in \mathcal{X}} f(\mathbf{x})$.

We aim for the iterative algorithm to find an approximate solution to Problem (1), which is defined as follows.

**Definition 2.4.** A point $\hat{\mathbf{x}}$ is called an $\epsilon$-suboptimal solution to Problem (1) if, for a given $\epsilon > 0$, it satisfies

$$f(\hat{\mathbf{x}}) - f(\mathbf{x}_{\star}) \leq \epsilon.$$

We also assume the stochastic component $F(\mathbf{x}; \boldsymbol{\xi})$ is convex and Lipschitz continuous with respect to $\mathbf{x}$.

**Assumption 2.5.** The stochastic component $F(\mathbf{x}; \boldsymbol{\xi})$ is convex in $\mathbf{x}$ for each fixed $\boldsymbol{\xi}$.

**Assumption 2.6.** There exists a constant $L \geq 0$ such that for all $\mathbf{x}, \mathbf{y} \in \mathbb{R}^d$, and any fixed $\boldsymbol{\xi}$, the following holds

$$\|F(\mathbf{x}; \boldsymbol{\xi}) - F(\mathbf{y}; \boldsymbol{\xi})\| \leq L\|\mathbf{x} - \mathbf{y}\|.$$

We further assume that the algorithm for solving Problem (1) has access to a stochastic zeroth-order oracle that returns unbiased stochastic function value estimates at two points.

**Assumption 2.7.** The stochastic zeroth-order oracle returns the stochastic evaluations $F(\mathbf{x}; \boldsymbol{\xi})$ and $F(\mathbf{y}; \boldsymbol{\xi})$ for given points $\mathbf{x} \in \mathbb{R}^d$ and $\mathbf{y} \in \mathbb{R}^d$, such that $\mathbb{E}_{\boldsymbol{\xi}}[F(\mathbf{x}; \boldsymbol{\xi})] = f(\mathbf{x})$ and $\mathbb{E}_{\boldsymbol{\xi}}[F(\mathbf{y}; \boldsymbol{\xi})] = f(\mathbf{y})$, where $\boldsymbol{\xi}$ is drawn from $\Xi$.

*Table 1.* We present the SZO complexity, step size $\eta_t$, and smoothing parameter $\mu_t$ for obtaining an $\epsilon$-suboptimal solution to Problem (1), where $t$ denotes the iteration index, $T$ is the total iteration budget, and $\bar{r}_t$ and $G_t$ are defined in Algorithm 1.

| Algorithms | Parameter-Free | SZO Complexity | $\eta_t$ | $\mu_t$ |
|---|---|---|---|---|
| RSNSO$^{\sharp}$ 
 Nesterov & Spokoiny (2017) | No | $\mathcal{O}\left(\dfrac{d^2 L^2 s_0^2}{\epsilon^2}\right)$ | $\dfrac{s_0}{dL\sqrt{T}}$ | $s_0\sqrt{\dfrac{d}{T}}$ |
| TPGE$^{\ddagger}$ 
 Duchi et al. (2015) | No | $\tilde{\mathcal{O}}\left(\dfrac{dL^2 D_{\mathcal{X}}^2}{\epsilon^2}\right)$ | $\dfrac{D_{\mathcal{X}}}{L\sqrt{d\log(2d)}t}$ | $\dfrac{D_{\mathcal{X}}}{t}$ and $\dfrac{D_{\mathcal{X}}}{d^2 t^2}$ |
| TPBCO 
 Shamir (2017) | No | $\mathcal{O}\left(\dfrac{dL^2 D_{\mathcal{X}}^2}{\epsilon^2}\right)$ | $\dfrac{D_{\mathcal{X}}}{L\sqrt{dT}}$ | $D_{\mathcal{X}}\sqrt{\dfrac{d}{T}}$ |
| POEM 
 Theorem 4.9 | **Yes** | $\tilde{\mathcal{O}}\left(\dfrac{dL^2 D_{\mathcal{X}}^2}{\epsilon^2}\right)$ | $\dfrac{\bar{r}_t}{\sqrt{G_t}}$ | $\bar{r}_t\sqrt{\dfrac{d}{t+1}}$ |
| Lower bound 
 Duchi et al. (2015) | – | $\Omega\left(\dfrac{dL^2 D_{\mathcal{X}}^2}{\epsilon^2}\right)$ | – | – |

$\sharp$  The RSNSO (Nesterov & Spokoiny, 2017) does not require the assumption of a bounded domain, as its complexity depends on the distance between the initial point $\mathbf{x}_0$ and the solution $\mathbf{x}_\star$, denoted by $s_0 = \|\mathbf{x}_0 - \mathbf{x}_\star\|$, rather than the diameter $D_{\mathcal{X}}$. We discuss the case without the bounded domain assumption in detail in Section 5.

$\ddagger$  The TPGE method (Duchi et al., 2015) employs two sequences of stochastic finite difference, each with its own smoothing parameter.

## 2.2. Randomized Smoothing

Randomized smoothing is a widely used technique in zeroth-order optimization, which constructs a smooth surrogate of the objective function by applying perturbations along random directions (Duchi et al., 2012a; Gasnikov et al., 2022; Shamir, 2017; Yousefian et al., 2012; Nesterov & Spokoiny, 2017; Lin et al., 2022). In this work, we specifically focus on randomized smoothing based on the uniform distribution over the unit ball (Duchi et al., 2012a; Gasnikov et al., 2022; Shamir, 2017). Formally, we define the smooth surrogate of the objective function $f(\mathbf{x})$ as

$$f_\mu(\mathbf{x}) \triangleq \mathbb{E}_{\mathbf{u}\sim\mathbb{U}(\mathbb{B}^d)}[f(\mathbf{x}+\mu\mathbf{u})],$$

where $\mu > 0$ is the smoothing parameter. The following lemma establishes that the surrogate $f_\mu(\mathbf{x})$ preserves the convexity, and that the approximation error between $f(\mathbf{x})$ and $f_\mu(\mathbf{x})$ can be bounded in terms of $\mu$ (Shamir, 2017).

**Lemma 2.8** (Shamir (2017, Lemma 8)). *Under Assumptions 2.5 and 2.6, the smooth surrogate $f_\mu(\mathbf{x})$ is convex and satisfies $|f_\mu(\mathbf{x}) - f(\mathbf{x})| \le L\mu$ for all $\mathbf{x} \in \mathbb{R}^d$.*

The next lemma demonstrates that the surrogate $f_\mu(\mathbf{x})$ is differentiable, regardless of whether $f(\mathbf{x})$ is differentiable. Moreover, it shows that the gradient of $f_\mu(\mathbf{x})$ can be expressed in the form of the finite difference.

**Lemma 2.9** (Flaxman et al. (2004, Lemma 3.4)). *For a continuous function $f : \mathbb{R}^d \to \mathbb{R}$, the gradient of its smooth surrogate $f_\mu$ is given by*

$$\nabla f_\mu(\mathbf{x}) = \mathbb{E}_{\mathbf{v}\sim\mathbb{U}(\mathbb{S}^{d-1})}\left[\frac{d}{2\mu}(f(\mathbf{x}+\mu\mathbf{v}) - f(\mathbf{x}-\mu\mathbf{v}))\mathbf{v}\right],$$

*where $\mathbf{x} \in \mathbb{R}^d$ and $\mu > 0$.*

Based on Lemma 2.9, we define the stochastic finite difference as follows

$$\mathbf{g}(\mathbf{x}, \mu; \mathbf{v}, \boldsymbol{\xi}) \triangleq \frac{d}{2\mu}(F(\mathbf{x}+\mu\mathbf{v}; \boldsymbol{\xi}) - F(\mathbf{x}-\mu\mathbf{v}; \boldsymbol{\xi}))\mathbf{v}, \quad (2)$$

where $\mathbf{x} \in \mathcal{X}$, $\mu > 0$, $\mathbf{v} \sim \mathbb{U}(\mathbb{S}^{d-1})$ and $\boldsymbol{\xi} \sim \Xi$. Under Assumption 2.7, the function evaluation $F(\mathbf{x}; \boldsymbol{\xi})$ returned by the stochastic zeroth-order oracle is an unbiased estimator of $f(\mathbf{x})$. Consequently, the stochastic finite difference $\mathbf{g}(\mathbf{x}, \mu; \mathbf{v}, \boldsymbol{\xi})$ is an unbiased estimator of $\nabla f_\mu(\mathbf{x})$, that is,

$$\mathbb{E}_{\mathbf{v}\sim\mathbb{U}(\mathbb{S}^{d-1}), \boldsymbol{\xi}\sim\Xi}[\mathbf{g}(\mathbf{x}, \mu; \mathbf{v}, \boldsymbol{\xi})] = \nabla f_\mu(\mathbf{x}).$$

## 3. Parameter-Free Stochastic Zeroth-Order Optimization

We propose POEM, a parameter-free stochastic zeroth-order method, as described in Algorithm 1. POEM is built on the framework of projected SGD, following the iterative scheme

$$\mathbf{x}_{t+1} = \Pi_{\mathcal{X}}(\mathbf{x}_t - \eta_t \mathbf{g}_t), \quad (3)$$

where $\eta_t > 0$ denotes the step size, and the finite difference

$$\mathbf{g}_t \triangleq \mathbf{g}(\mathbf{x}_t, \mu_t; \mathbf{v}_t, \boldsymbol{\xi}_t) \quad (4)$$

is defined as in equation (2), with the smoothing parameter $\mu_t > 0$, and random variables $\mathbf{v}_t \sim \mathbb{U}(\mathbb{S}^{d-1})$ and $\boldsymbol{\xi}_t \sim \Xi$.

We aim to make both the step size $\eta_t$ and the smoothing parameter $\mu_t$ in equations (3) and (4) tuning-free, and still achieve near-optimal convergence rates. This is more challenging than existing stochastic parameter-free first-order

---

**Algorithm 1** POEM

1: **Input:** $\mathbf{x}_0 \in \mathcal{X}, \quad r_\epsilon \in (0, D_\mathcal{X}], \quad T \geq 1$

2: $\bar{r}_{-1} = r_\epsilon, \quad G_{-1} = 0$

3: **for** $t = 0, \ldots, T-1$ **do**

4: $\quad \bar{r}_t = \max\{\bar{r}_{t-1}, \|\mathbf{x}_t - \mathbf{x}_0\|\}$

5: $\quad \mu_t = \bar{r}_t \sqrt{\dfrac{d}{t+1}}$

6: $\quad \mathbf{v}_t \sim \mathbb{U}(\mathbb{S}^{d-1}), \ \boldsymbol{\xi}_t \sim \Xi$

7: $\quad \mathbf{g}_t = \dfrac{d}{2\mu_t}(F(\mathbf{x}_t + \mu_t\mathbf{v}_t; \boldsymbol{\xi}_t) - F(\mathbf{x}_t - \mu_t\mathbf{v}_t; \boldsymbol{\xi}_t))\mathbf{v}_t$

8: $\quad G_t = G_{t-1} + \|\mathbf{g}_t\|^2$

9: $\quad \eta_t = \dfrac{\bar{r}_t}{\sqrt{G_t}}$

10: $\quad \mathbf{x}_{t+1} = \Pi_\mathcal{X}(\mathbf{x}_t - \eta_t\mathbf{g}_t)$

11: **end for**

12: **Output:** $\bar{\mathbf{x}}_{\tau_T}$ where $\tau_T \triangleq \arg\max_{t \leq T} \sum_{k=0}^{t-1} \dfrac{\bar{r}_k}{\bar{r}_t}$

---

methods, which focus only on adapting the step size. Inspired by the strategy of DoG (Ivgi et al., 2023a), we schedule the step size $\eta_t$ based on the ratio between the distance from the initial point and the norm of the stochastic finite difference. Specifically, we define the cumulative squared gradient norm as $G_t \triangleq \sum_{k=0}^{t} \|\mathbf{g}_k\|^2$, the distance to the initial point as $r_t \triangleq \|\mathbf{x}_t - \mathbf{x}_0\|$, and the maximum distance as $\bar{r}_t \triangleq \max_{k \leq t} r_k \vee r_\epsilon$, where $r_\epsilon > 0$ denotes the initial movement. We define the step size at the $t$-th iteration as

$$\eta_t \triangleq \frac{\bar{r}_t}{\sqrt{G_t}}. \tag{5}$$

For initialization, we require the movement $r_\epsilon \in (0, D_\mathcal{X}]$. As we will show in Sections 4 and 6, the choice of $r_\epsilon$ influences the theoretical convergence rates only by a logarithmic term and has minimal impact on practical performance.

Moreover, we define the smoothing parameter as

$$\mu_t \triangleq \bar{r}_t \sqrt{\frac{d}{t+1}}, \tag{6}$$

which is adaptive and generally larger than those used in existing stochastic zeroth-order methods (Nesterov & Spokoiny, 2017; Shamir, 2017; Duchi et al., 2015; Ghadimi & Lan, 2013; Duchi et al., 2012b; Rando et al., 2024). For example, Nesterov & Spokoiny (2017); Shamir (2017) set $\mu_t = \mathcal{O}(\sqrt{d/T})$ in their analysis, depending on the total iteration budget $T$; Duchi et al. (2015) uses $\mu_t = \mathcal{O}(1/(dt)^2)$, which may be quite small in high-dimensional settings. Recall that the smoothing parameter $\mu_t$ appears in the denominator of the stochastic finite difference in equation (2). Consequently, a larger $\mu_t$ improves numerical stability.

## 4. The Complexity Analysis

In this section, we show that POEM (Algorithm 1) achieves near-optimal SZO complexity. The detailed proofs of the results presented here are provided in Appendix B.

Our analysis focuses on the weighted average of the iterates generated by POEM, defined as

$$\bar{\mathbf{x}}_t \triangleq \frac{1}{\sum_{k=0}^{t-1} \bar{r}_k} \sum_{k=0}^{t-1} \bar{r}_k\mathbf{x}_k. \tag{7}$$

Since the objective function $f$ is convex (Assumption 2.5), we apply Jensen's inequality to bound the optimality gap

$$f(\bar{\mathbf{x}}_t) - f(\mathbf{x}_\star) \leq \frac{1}{\sum_{k=0}^{t-1} \bar{r}_k} \sum_{k=0}^{t-1} \bar{r}_k(f(\mathbf{x}_k) - f(\mathbf{x}_\star)). \tag{8}$$

By combining inequality (8) with Lemma 2.8, we obtain

$$
\begin{aligned}
&f(\bar{\mathbf{x}}_t) - f(\mathbf{x}_\star) \\
&\leq \frac{1}{\sum_{k=0}^{t-1} \bar{r}_k} \sum_{k=0}^{t-1} \bar{r}_k(f_{\mu_k}(\mathbf{x}_k) - f_{\mu_k}(\mathbf{x}_\star) + 2L\mu_k) \\
&\leq \frac{1}{\sum_{k=0}^{t-1} \bar{r}_k} \sum_{k=0}^{t-1} \bar{r}_k(\langle \nabla f_{\mu_k}(\mathbf{x}_k), \mathbf{x}_k - \mathbf{x}_\star \rangle + 2L\mu_k),
\end{aligned}
\tag{9}
$$

where the last inequality follows from the convexity of $f_{\mu_k}$. We decompose the sum in the final line of equation (9) into

$$\underbrace{\sum_{k=0}^{t-1} \bar{r}_k\langle \mathbf{g}_k, \mathbf{x}_k - \mathbf{x}_\star \rangle}_{\text{the weighted regret}} + \underbrace{\sum_{k=0}^{t-1} \bar{r}_k\langle \boldsymbol{\Delta}_k, \mathbf{x}_k - \mathbf{x}_\star \rangle}_{\text{the noise from } \mathbf{g}_k} + \underbrace{\sum_{k=0}^{t-1} 2L\bar{r}_k\mu_k}_{\text{the noise from } \mu_k}, \tag{10}$$

where $\boldsymbol{\Delta}_k \triangleq \nabla f_{\mu_k}(\mathbf{x}_k) - \mathbf{g}_k$. The regret term is a standard component in the complexity analysis of stochastic zeroth-order and first-order methods (Shalev-Shwartz, 2012; Duchi et al., 2015; Balasubramanian & Ghadimi, 2018; Ivgi et al., 2023a). In the context of POEM, this term is scaled by the weights $\{\bar{r}_k\}_{k=0}^{t-1}$, requiring careful control. The noise from $\mathbf{g}_k$ arises from the discrepancy between the true gradient $\nabla f_{\mu_k}$ and its unbiased estimator $\mathbf{g}_k$. The noise from $\mu_k$ reflects the approximation error between the objective function $f$ and its smooth surrogate $f_{\mu_k}$. Notably, this noise doesn't appear in the analysis of first-order methods.

We now present upper bounds for the three components in equation (9): the weighted regret term, the noise from $\mathbf{g}_k$, and the noise from $\mu_k$. To facilitate the analysis, we define

$$s_t \triangleq \|\mathbf{x}_t - \mathbf{x}_\star\| \qquad \text{and} \qquad \bar{s}_t \triangleq \max_{k \leq t} s_k.$$

The following lemma provides an upper bound on the weighted regret for SGD-type iterations. Notably, this result holds independently of the specific form of the gradient estimator or the choice of step size, and thus applies directly to POEM (Algorithm 1).

**Lemma 4.1** (Ivgi et al. (2023a, Lemma 3.4)). *For the iteration scheme (3), the weighted regret satisfies*

$$\sum_{k=0}^{t-1} \bar{r}_k \langle \mathbf{g}_k, \mathbf{x}_k - \mathbf{x}_\star \rangle \leq \bar{r}_t \left( 2\bar{s}_t + \bar{r}_t \right) \sqrt{G_{t-1}}.$$

To analyze the noise from $\mathbf{g}_k$, we first establish upper bounds for $\|\mathbf{g}_t\|$ and its second moment $\mathbb{E}[\|\mathbf{g}_t\|^2]$.

**Lemma 4.2** (Shamir (2017, Lemma 10)). *Under Assumption 2.6, POEM (Algorithm 1) satisfies*

$$\|\mathbf{g}_t\| \leq Ld \quad and \quad \mathbb{E}[\|\mathbf{g}_t\|^2] \leq cL^2 d,$$

*where $c > 0$ is a numerical constant.*

*Remark* 4.3. This lemma provides $\mathcal{O}(d)$ upper bounds for both $\|\mathbf{g}_t\|$ and $\mathbb{E}[\|\mathbf{g}_t\|^2]$, which are crucial for achieving optimal dependence on the dimension $d$ in the convergence rates. Unlike the original proof in Shamir (2017, Lemma 10), our analysis is based on the Euclidean norm and offers a more concise argument by avoiding the use of fourth-order moments of the gradient estimator (see Appendix B.1).

Based on Lemma 4.2, we can provide an upper bound for the term $\|\boldsymbol{\Delta}_k\| = \|\nabla f_{\mu_k}(\mathbf{x}_k) - \mathbf{g}_k\|$. We then use a concentration inequality for martingale differences (Howard et al., 2021; Ivgi et al., 2023b) to control the noise from $\mathbf{g}_k$.

**Lemma 4.4.** *Under Assumptions 2.6 and 2.7, for any $\delta \in (0,1)$, POEM (Algorithm 1) satisfies*

$$\mathbb{P}\left( \exists t \leq T : \left| \sum_{k=0}^{t-1} \bar{r}_k \langle \boldsymbol{\Delta}_k, \mathbf{x}_k - \mathbf{x}_\star \rangle \right| \geq b_t \right) \leq \delta,$$

*where we define $b_t \triangleq 8\bar{r}_{t-1}\bar{s}_{t-1}\sqrt{\theta_{t,\delta}G_{t-1} + 4L^2d^2\theta_{t,\delta}^2}$, and $\theta_{t,\delta} \triangleq \log(60\log(6t/\delta))$.*

Next, we provide an upper bound for the noise from $\mu_k$.

**Lemma 4.5.** *POEM (Algorithm 1) satisfies*

$$\sum_{k=0}^{t-1} 2L\bar{r}_k\mu_k \leq 4L\bar{r}_{t-1}^2\sqrt{dt},$$

*where $\log_+(\cdot) \triangleq \log(\cdot) + 1$.*

*Remark* 4.6. The choice of the smoothing parameter in (6) implies that $\mu_k = \mathcal{O}(\sqrt{d/k})$. Using this expression, we can bound the series by approximating it with an integral, which leads to the stated result in Lemma 4.5. A detailed proof is provided in Appendix B.3.

By combining Lemmas 4.1, 4.4, and 4.5 with equations (9) and (10), we obtain the following upper bound on the optimality function value gap.

**Proposition 4.7.** *Under Assumptions 2.5, 2.6 and 2.7, for any $\delta \in (0,1)$ and $t \in \mathbb{N}_+$, POEM (Algorithm 1) satisfies*

$$f(\bar{\mathbf{x}}_t) - f(\mathbf{x}_\star) \leq \frac{16\theta_{t,\delta}(\bar{r}_t + s_0)(\sqrt{G_{t-1}} + Ld + L\sqrt{dt})}{\sum_{k=0}^{t-1} \bar{r}_k/\bar{r}_t},$$

*with probability at least $1 - \delta$, where $\theta_{t,\delta} = \log(60\log(t/\delta))$.*

We then consider a lower bound for $\sum_{k=0}^{t-1} \bar{r}_k/\bar{r}_t$. To this end, we introduce the following key lemma.

**Lemma 4.8** (Ivgi et al. (2023a, Lemma 3.7)). *Let $a_0, a_1, \ldots, a_T$ be a positive non-decreasing sequence, then*

$$\max_{t \leq T} \sum_{i < t} \frac{a_i}{a_t} \geq \frac{1}{\mathrm{e}} \left( \frac{T}{\log_+(a_T/a_0)} - 1 \right),$$

*where $T \in \mathbb{N}_+$.*

Since the sequence $\{\bar{r}_t\}_{t=0}^T$ is positive and non-decreasing, we can apply Lemma 4.8 with $a_t = \bar{r}_t$. This yields

$$\max_{t \leq T} \sum_{k=0}^{t-1} \frac{\bar{r}_k}{\bar{r}_t} = \sum_{k=0}^{\tau_T - 1} \frac{\bar{r}_k}{\bar{r}_{\tau_T}} \geq \Omega\left( \frac{T}{\log_+(\bar{r}_{\tau_T}/r_\epsilon)} \right), \quad (11)$$

where $\tau_T \triangleq \arg\max_{1 \leq t \leq T} \sum_{k=0}^{t-1} \bar{r}_k/\bar{r}_t$.

Before stating the main result, we define the probability space $(\Omega_0, \mathcal{F}_0, \mathbb{P})$, where $\Omega_0$ denotes the sample space associated with POEM (Algorithm 1) for a given $\mathbf{x}_0$ and $r_\epsilon$, $\mathcal{F}_0$ is the sigma field generated by the random variable sequences $\{\mathbf{v}_t\}_{t=0}^{T-1}$ and $\{\boldsymbol{\xi}_t\}_{t=0}^{T-1}$, and $\mathbb{P}$ is a probability measure defined on $\mathcal{F}_0$. Next, we define the event

$$\Omega_\delta \triangleq \left\{ \omega \in \Omega_0 : \forall t \leq T, \left| \sum_{k=0}^{t-1} \bar{r}_k \langle \boldsymbol{\Delta}_k, \mathbf{x}_k - \mathbf{x}_\star \rangle \right| < b_t \right\}.$$

By Lemma 4.4, given $\delta \in (0,1)$, we have $\mathbb{P}(\Omega_\delta) \geq 1 - \delta$. We then define the sigma field $\mathcal{F}_\delta \triangleq \{A : A \subset \Omega_\delta\} \cap \mathcal{F}_0$, which satisfies $\mathcal{F}_\delta \subset \mathcal{F}_0$. Furthermore, the Lipschitz continuity of $f$ and the boundedness of the domain $\mathcal{X}$, ensure that $\mathbb{E}|f(\bar{\mathbf{x}}_{\tau_T}) - f(\mathbf{x}_\star)| \leq LD_\mathcal{X} < \infty$. Therefore, the conditional expectation $\mathbb{E}[f(\bar{\mathbf{x}}_{\tau_T}) - f(\mathbf{x}_\star) \mid \mathcal{F}_\delta]$ exists and is unique (Durrett, 2019, Chapter 4.1).

By combining Lemma 4.2, Proposition 4.7, and equation (11), we arrive at the main result.

**Theorem 4.9.** *Under Assumptions 2.1, 2.5, 2.6, and 2.7, for any $\delta \in (0,1)$ and $T \in \mathbb{N}_+$, POEM (Algorithm 1) initialized with $\mathbf{x}_0 \in \mathcal{X}$ and $r_\epsilon \in (0, D_\mathcal{X}]$ satisfies*

$$\mathbb{E}[f(\bar{\mathbf{x}}_{\tau_T}) - f(\mathbf{x}_\star) \mid \mathcal{F}_\delta]$$
$$\leq \mathcal{O}\left( \left( \frac{d}{T} + \frac{\sqrt{d}}{\sqrt{T}} \right) \theta_{T,\delta} LD_\mathcal{X} \log_+\left( \frac{D_\mathcal{X}}{r_\epsilon} \right) \right),$$

*with probability at least $1 - \delta$, where $\theta_{T,\delta} \triangleq \log(60\log(6T/\delta))$.*

*By suppressing the logarithmic factors using the $\tilde{\mathcal{O}}(\cdot)$ notation, the SZO complexity for finding a conditionally expected $\epsilon$-suboptimal solution with probability at least $1 - \delta$ is*

$$\tilde{\mathcal{O}}\left(\frac{dL^2 D_{\mathcal{X}}^2}{\epsilon^2}\right).$$

The SZO complexity established in Theorem 4.9 matches the lower bound for stochastic zeroth-order optimization established by Duchi et al. (2015).

*Remark* 4.10. As $\delta \to 0$, $\Omega_\delta \to \Omega_0$ in probability and $\mathcal{F}_\delta$ approaches $\mathcal{F}_0$, which implies $\mathbb{E}[f(\bar{\mathbf{x}}_{\tau_T}) - f(\mathbf{x}_\star) \mid \mathcal{F}_\delta]$ approaches $\mathbb{E}[f(\bar{\mathbf{x}}_{\tau_T}) - f(\mathbf{x}_\star) \mid \mathcal{F}_0] = f(\bar{\mathbf{x}}_{\tau_T}) - f(\mathbf{x}_\star)$.

## 5. Results for Unbounded Domains

In this section, we extend our method to address the stochastic convex optimization problem in settings where the domain may be unbounded. To accommodate this, we relax Assumption 2.1 as follows.

**Assumption 5.1.** The domain $\mathcal{X} \subseteq \mathbb{R}^d$ is closed and convex. Moreover, there exists a point $\mathbf{x}_\star \in \mathcal{X}$ such that $f(\mathbf{x}_\star) = \min_{\mathbf{x} \in \mathcal{X}} f(\mathbf{x})$.

*Remark* 5.2. In our analysis for the bounded domain (Theorem 4.9), the upper bound on $\mathbb{E}[f(\bar{\mathbf{x}}_{\tau_T}) - f(\mathbf{x}_\star) \mid \mathcal{F}_\delta]$ includes the term $\log_+(D_{\mathcal{X}}/r_\epsilon)$, where $r_\epsilon \in (0, D_{\mathcal{X}}]$. However, this term may become invalid after relaxing Assumption 2.1 to Assumption 5.1, as $D_{\mathcal{X}}$ could be infinite.

In the remainder of this section, we first modify POEM by introducing an overestimate of the Lipschitz constant $L$ to address the problem without assuming a bounded domain. We then show that such estimation is unavoidable in the unbounded setting. The detailed proofs for the results presented in this section are deferred to Appendix C.

We introduce the quantity

$$G'_t \triangleq 8^4 \theta_{T,\delta} \log_+^2(t+2)(G_{t-1} + 16\theta_{T,\delta} d^2 \bar{L}^2), \quad (12)$$

where $\theta_{T,\delta} = \log(60 \log(6T/\delta))$, $G_t = \sum_{k=0}^t \|\mathbf{g}_k\|^2$ as defined in Section 3 and $\bar{L}$ is an overestimate of the Lipschitz constant $L$ such that $\bar{L} \geq L$.

For the unbounded domain, we modify POEM (Algorithm 1) by updating the step size and smoothing parameter as

$$\eta_t = \frac{\bar{r}_t}{\sqrt{G'_t}} \qquad \text{and} \qquad \mu_t = \frac{d\bar{r}_t}{(t+1)^2}, \quad (13)$$

where $r_t = \|\mathbf{x}_t - \mathbf{x}_0\|$ and $\bar{r}_t = \max_{k \leq t} r_k \vee r_\epsilon$, following the notation in Section 3. We also define $G_{-1} = 0$ for equation (12) when $t = 0$. Note that the parameters $T$ and $\delta$ only influence the logarithmic factor in $G'_t$. Furthermore, the term $16\theta_{T,\delta} d^2 \bar{L}^2$ in equation (12) becomes relatively insignificant compared to $G_{t-1}$ as $t$ grows large.

We now provide the complexity analysis for the modified POEM. Unlike in the bounded domain setting, the quantity $\bar{r}_t$ cannot be simply controlled via $D_{\mathcal{X}}$. Instead, our goal is to show that $\bar{r}_t = \mathcal{O}(s_0)$, which implies that $\mathbf{x}_t$ remains close to both $\mathbf{x}_0$ and $\mathbf{x}_\star$. Starting from the iteration update $\mathbf{x}_{k+1} = \Pi_{\mathcal{X}}(\mathbf{x}_k - \eta_k \mathbf{g}_k)$, we obtain the inequality

$$\|\mathbf{x}_{k+1} - \mathbf{x}_\star\|^2 \leq \|\mathbf{x}_k - \mathbf{x}_\star - \eta_k \mathbf{g}_k\|^2.$$

Rewriting this inequality using the definition of $s_k$ in Section 3, we have

$$s_{k+1}^2 - s_k^2 \leq \eta_k^2 \|\mathbf{g}_k\|^2 + 2\eta_k \langle \boldsymbol{\Delta}_k, \mathbf{x}_k - \mathbf{x}_\star \rangle - 2\eta_k \langle \nabla f_{\mu_k}(\mathbf{x}_k), \mathbf{x}_k - \mathbf{x}_\star \rangle,$$

where $\boldsymbol{\Delta}_k = \nabla f_{\mu_k}(\mathbf{x}_k) - \mathbf{g}_k$. Summing the above inequality over $k = 0, 1, \ldots, t-1$, we obtain

$$
\begin{aligned}
s_t^2 - s_0^2 \leq &\sum_{k=0}^{t-1} \eta_k^2 \|\mathbf{g}_k\|^2 + 2\sum_{k=0}^{t-1} \eta_k \langle \boldsymbol{\Delta}_k, \mathbf{x}_k - \mathbf{x}_\star \rangle \\
&+ 2\sum_{k=0}^{t-1} \eta_k \langle \nabla f_{\mu_k}(\mathbf{x}_k), \mathbf{x}_\star - \mathbf{x}_k \rangle.
\end{aligned}
\quad (14)
$$

Therefore, we can upper bound $\bar{r}_t$ by controlling each of the three terms on the right-hand side of inequality (14). Note that the last term in equation (14) does not appear in the analysis of first-order methods like DoG (Ivgi et al., 2023a). Following the analysis in Appendix C.1, we establish the follow upper bound for $\bar{r}_t$.

**Proposition 5.3.** *For any $\delta \in (0, 1)$, POEM (Algorithm 1), with settings $\eta_t = \bar{r}_t/\sqrt{G'_t}$, $\mu_t = d\bar{r}_t/(t+1)^2$, and $r_\epsilon \in (0, 3s_0]$, satisfies $\tilde{\mathbb{P}}(\bar{r}_T > 3s_0) \leq \delta$.*

We consider the probability space $(\tilde{\Omega}_0, \tilde{\mathcal{F}}_0, \tilde{\mathbb{P}})$, where $\tilde{\Omega}_0$ is the sample space of the Algorithm 1 under the modified settings used in Proposition 5.3, $\tilde{\mathcal{F}}_0$ is the sigma field generated by the random sequences $\{\mathbf{v}_t\}_{t=0}^{T-1}$ and $\{\boldsymbol{\xi}_t\}_{t=0}^{T-1}$, and $\tilde{\mathbb{P}}$ is a probability measure defined on $\tilde{\mathcal{F}}_0$.

Following the settings of Proposition 5.3, we define the set

$$\tilde{\Omega}_\delta \triangleq \left\{ \omega \in \tilde{\Omega}_0 : \forall t \leq T, \left| \sum_{k=0}^{t-1} \tilde{\eta}_k \langle \boldsymbol{\Delta}_k, \mathbf{x}_k - \mathbf{x}_\star \rangle \right| \leq s_0^2 \right\},$$

where $\tilde{\eta}_t \triangleq \eta_t \cdot \mathbb{I}(t < \zeta)$ and $\zeta \triangleq \min\{t \in \mathbb{N} \mid \bar{r}_t > 3s_0\}$.

The derivation of Proposition 5.3 (see Appendix C.1) shows that if $r_\epsilon \leq 3s_0$, then $\bar{r}_T \leq 3s_0$ for all $\omega \in \tilde{\Omega}_\delta$. Moreover, it holds that $\tilde{\mathbb{P}}(\tilde{\Omega}_\delta) \geq 1 - \delta$.

Similar to Proposition 4.7, we can establish an upper bound on the optimality gap for the unbounded setting as follows (see Appendix C.2 for the proof).

**Proposition 5.4.** *Under Assumptions 2.5, 2.6 and 2.7, for any $\delta \in (0, 1)$, POEM (Algorithm 1) with the modified*

*settings from Proposition 5.3 satisfies*

$$f(\bar{\mathbf{x}}_t) - f(\mathbf{x}_\star) \leq \frac{20\theta_{t,\delta}(\bar{r}_t + s_0)(\sqrt{G'_{t-1}} + Ld)}{\sum_{k=0}^{t-1} \bar{r}_k/\bar{r}_t}, \quad (15)$$

*with probability at least* $1 - \delta$, *where* $\theta_{t,\delta} = \log(60 \log(t/\delta))$.

Following the settings of Proposition 5.4, we define the set

$$\hat{\Omega}_\delta \triangleq \left\{ \omega \in \tilde{\Omega}_0 : \forall t \leq T, \left| \sum_{k=0}^{t-1} \bar{r}_k \langle \boldsymbol{\Delta}_k, \mathbf{x}_k - \mathbf{x}_\star \rangle \right| < b_t \right\},$$

where $b_t = 8\bar{r}_{t-1}\bar{s}_{t-1}\sqrt{\theta_{t,\delta}G_{t-1} + 4L^2d^2\theta_{t,\delta}^2}$. The proof of Proposition 5.4 shows that inequality (15) holds for all $\omega \in \hat{\Omega}_\delta$. Moreover, we have $\tilde{\mathbb{P}}(\hat{\Omega}_\delta) \geq 1 - \delta$.

Next, we define $\tilde{F}_\delta \triangleq \{A : A \subset \tilde{\Omega}_\delta \cap \hat{\Omega}_\delta\} \cap \tilde{\mathcal{F}}_0$, which is a sub-sigma-field of $\tilde{\mathcal{F}}_0$. Since the probability space is constructed for an algorithm with finite $T$, we can conclude the conditional expectation $\mathbb{E}[f(\bar{\mathbf{x}}_{\tau_T}) - f(\mathbf{x}_\star) \mid \tilde{\mathcal{F}}_\delta]$ exists and is unique, even if the domain $D_\mathcal{X}$ is unbounded. (Please see Appendix C.3 for the detailed derivation.)

We now combine Propositions 5.3 and 5.4 to establish a convergence result without a bounded domain assumption.

**Theorem 5.5.** *Under Assumptions 2.5, 2.6, 2.7, and 5.1, for any* $\delta \in (0, 1/2)$, *POEM (Algorithm 1) with the modified settings used in Proposition 5.3, satisfies*

$$\mathbb{E}[f(\bar{\mathbf{x}}_{\tau_T}) - f(\mathbf{x}_\star) \mid \tilde{\mathcal{F}}_\delta]$$
$$\leq \mathcal{O}\left( \left( \frac{d(L + \bar{L})}{T} + \frac{\sqrt{d}L}{\sqrt{T}} \right) \alpha_{T,\delta} s_0 \log_+ \left( \frac{s_0}{r_\epsilon} \right) \right)$$

*with probability at least* $1 - 2\delta$, *where* $s_0 = \|\mathbf{x}_0 - \mathbf{x}_\star\|$ *and* $\alpha_{T,\delta} \triangleq \log_+(T + 1) \log(60 \log(T/\delta))$.

By setting $\bar{L} = L$, the upper bound on the conditional expectation shown in Theorem 5.5 simplifies to

$$\mathcal{O}\left( \left( \frac{d}{T} + \frac{\sqrt{d}}{\sqrt{T}} \right) \alpha_{T,\delta} L s_0 \log_+ \left( \frac{s_0}{r_\epsilon} \right) \right).$$

This yields the SZO complexity of $\tilde{\mathcal{O}}(dL^2 s_0^2/\epsilon^2)$ for finding an $\epsilon$-suboptimal solution $\bar{\mathbf{x}}_{\tau_T}$, which improves upon the $\mathcal{O}(d^2 L^2 s_0^2/\epsilon^2)$ complexity established by Nesterov & Spokoiny (2017).

However, the settings in Theorem 5.5 (also Proposition 5.3) require that $r_\epsilon \in (0, 3s_0]$, where $s_0 = \|\mathbf{x}_0 - \mathbf{x}_\star\|$ is unknown in practice. Furthermore, the first term in the upper bound of Theorem 5.5 depends linearly on $\bar{L}$. Ideally, we would like to design an algorithm that achieves an SZO complexity close to $\tilde{\mathcal{O}}(dL^2 s_0^2/\epsilon^2)$, with only logarithmic dependence on uncertain problem parameters such as $r_\epsilon$ and $\bar{L}$. Unfortunately, we show that such an ideal, fully

parameter-free zeroth-order algorithm for stochastic convex optimization without a bounded domain assumption is provably unattainable.

We assume that the stochastic zeroth-order algorithm $\mathcal{A}$ accepts valid estimates $\bar{L}$, $\underline{L}$, $\bar{s}$ and $\underline{s}$ such that $\underline{L} \leq L \leq \bar{L}$ and $\underline{s} \leq s_0 \leq \bar{s}$. Based on this setup, we establish the following lower bound on the function value gap for the stochastic convex optimization problem.

**Theorem 5.6.** *Let* $\theta : \mathbb{R}^4 \to \mathbb{R}$ *be any polylogarithmic function, let* $d \in \mathbb{N}$, *and let* $\mathcal{A}$ *be a stochastic zeroth-order algorithm satisfying Assumption 2.7, with valid estimates* $\underline{L}$, $\bar{L}$, $\underline{s}$, *and* $\bar{s}$. *Then, there exists an* $L$-Lipschitz convex function $f : \mathbb{R}^d \to \mathbb{R}$ *such that, for any initial point* $\mathbf{x}_0 \in \mathbb{R}^d$ *and any number of SZO calls* $T \geq 2$, *the algorithm* $\mathcal{A}$ *returns a point* $\hat{\mathbf{x}}$ *satisfying*

$$f(\hat{\mathbf{x}}) - f_\star > \theta\left( \frac{\bar{L}}{\underline{L}}, \frac{\bar{s}}{\underline{s}}, T, d \right) \cdot \frac{\sqrt{d}Ls_0}{\sqrt{T}}$$

*with probability at least* $1/e$.

*Remark 5.7.* In a recent work, Khaled & Jin (2024) showed the impossibility of an ideal parameter-free algorithm for stochastic first-order optimization by constructing a hard instance in the one-dimensional setting. In contrast, the lower bound for zeroth-order optimization established in Theorem 5.6 must additionally consider the dependence on the dimension of the problem, highlighting a key distinction from the first-order case.

# 6. Numerical Experiments

This section presents numerical experiments to evaluate the empirical performance of POEM (Algorithm 1). We consider a stochastic optimization problem of the form

$$\min_{\mathbf{x} \in \mathcal{X}} f(\mathbf{x}) \triangleq \mathbb{E}_{(\mathbf{a},b)}[F(\mathbf{x}; \mathbf{a}, b)],$$

where $F(\mathbf{x}; \mathbf{a}, b) = \max\{0, 1 - b\mathbf{a}^\top \mathbf{x}\}$, $(\mathbf{a}, b) \in \mathbb{R}^d \times \{\pm 1\}$ is uniformly sampled from a binary classification dataset $\{(\mathbf{a}_i, b_i)\}_{i=1}^n$. The feasible set is defined as $\mathcal{X} = \{\mathbf{x} \in \mathbb{R}^d : \|\mathbf{x}\| \leq R\}$ with radius $R = 1$. We conduct experiments on benchmark datasets from (Chang & Lin, 2011), including "mushrooms" ($d = 112$, $n = 8124$), "a9a" ($d = 123$, $n = 32,561$), and "w8a" ($d = 300$, $n = 49,749$). For comparison, we consider two stochastic zeroth-order algorithms: Two-Point Gradient Estimates (TPGE) method (Duchi et al., 2015) and Two-Point Bandit Convex Optimization (TPBCO) method (Shamir, 2017).

Figure 1 shows the comparison of SZO complexity versus function values. For POEM, the initial movement is set to $r_\epsilon = 10^{-2}$. Baseline methods are evaluated under two configurations: theoretical parameter settings (TPGE-T, TPBCO-T), and well-tuned step sizes (TPGE-E, TPBCO-E). Results demonstrate that POEM converges faster than

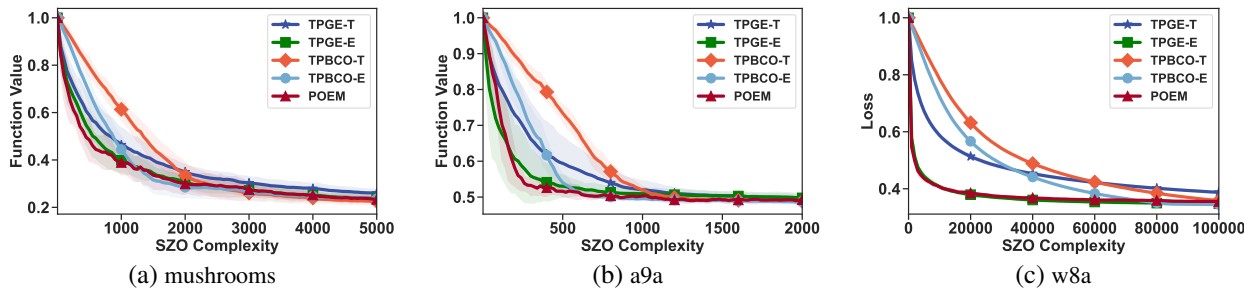

*Figure 1.* The comparison on the SZO complexity versus the function value during the iterations.

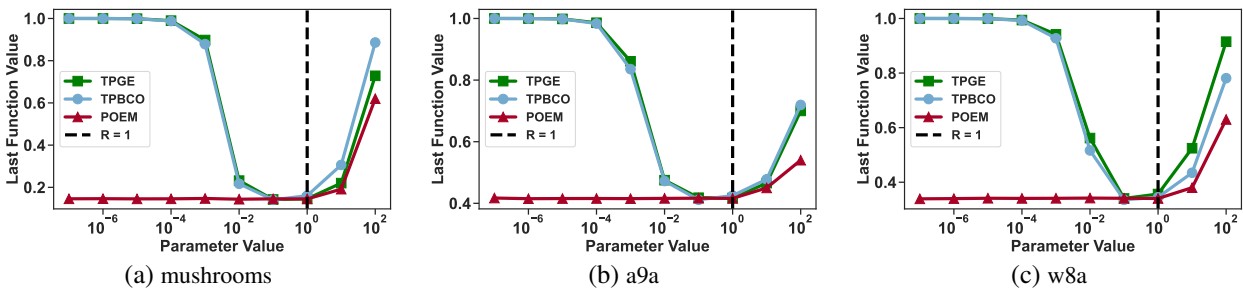

*Figure 2.* The comparison on the parameter settings ($r_\epsilon$ for POEM and $1/L$ for other methods) against $f(\mathbf{x}_T)$.

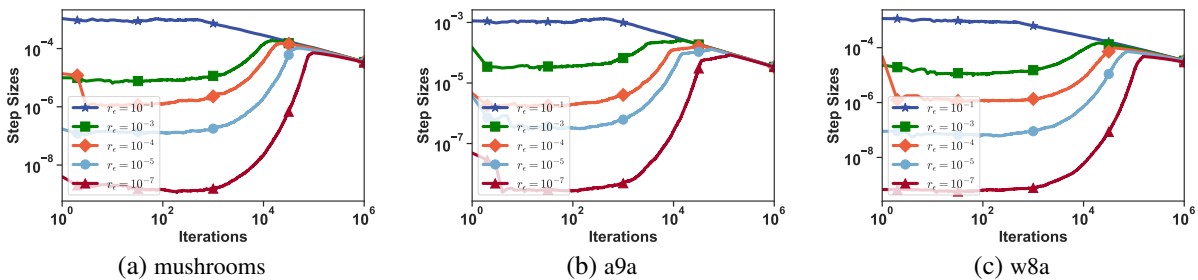

*Figure 3.* The change of the step size with difference $r_\epsilon$ for POEM.

TPGE-T and TPBCO-T, while its performance is comparable to the well-tuned variants (TPGE-E, TPBCO-E).

We further study the practical impact of parameter settings. Specifically, we present the objective function value at the final iteration ($T = 10^6$) across all algorithms under different configurations. Figure 2 summarizes these results, where we tune the initial movement $r_\epsilon$ in POEM and the term $1/L$ in baseline methods over the range $\{10^{-7}, 10^{-6}, \ldots, 10^2\}$. It is clear that POEM exhibits greater stability across parameter settings compared to baseline methods. More importantly, when $r_\epsilon \leq R = 1$, the choice of $r_\epsilon$ has negligible impact on the function value. This supports our theoretical analysis (Theorem 4.9), where $r_\epsilon$ only influences the logarithmic term in the complexity bound if it does not exceed the domain diameter. Additionally, we track the evolution of step sizes in POEM under varying $r_\epsilon$. As shown in Figure 2, the step sizes converge to similar values across all configurations, highlighting the algorithm's adaptive nature.

## 7. Conclusion

In this paper, we propose POEM, a novel zeroth-order optimization algorithm for stochastic convex optimization. It can dynamically schedule both the step size and the smoothing parameter during iterations. We show that POEM achieves near-optimal stochastic zeroth-order oracle complexity for problems with bounded domains. Notably, its initialization only impacts convergence rates by a logarithmic factor. We further extend POEM to unbounded domains and derive a lower bound, which reveals that an ideal parameter-free algorithm is impossible in such settings. We also conduct numerical experiments to confirm the practical efficiency of POEM.

In future work, we are interested in extending the ideas of POEM to broader applications, including zeroth-order optimization for minimax and bilevel problems. Another promising direction is the development of parameter-free zeroth-order methods for finite-sum optimization.

## Acknowledgements

This work is supported by the National Natural Science Foundation of China (No. 62206058), the Major Key Project of PCL under Grant PCL2024A06, and Shanghai Basic Research Program (23JC1401000).

## Impact Statement

This paper presents work whose goal is to advance the field of Machine Learning. There are many potential societal consequences of our work, none which we feel must be specifically highlighted here.

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

# A. Some Basic Results

We first present some basic lemmas.

**Lemma A.1** (Shamir (2017, Lemma 9)). *Suppose* $\mathbf{v} \sim \mathbb{U}(\mathbb{S}^{d-1})$. *Then, for any function* $h : \mathbb{R}^d \to \mathbb{R}$ *hat is L-Lipschitz with respect to the* $\ell_2$-*norm, the following concentration inequality holds*

$$\mathbb{P}(|h(\mathbf{v}) - \mathbb{E}_{\mathbf{v} \sim \mathbb{U}(\mathbb{S}^{d-1})}[h(\mathbf{v})]| \geq t) \leq 2 \exp\left(-\frac{cdt^2}{L^2}\right),$$

*where c is a numerical constant.*

**Lemma A.2** (Ivgi et al. (2023a, Lemma D.2)). *Let S be the set of non-negative and non-decreasing sequences. Let* $c > 0$ *and let* $X_t$ *be a martingale difference sequence adapted to* $\mathcal{F}_t$ *such that* $|X_t| \leq c$ *with probability 1 for all t . Then, for all* $\delta \in (0, 1)$ *and* $\hat{X}_t \in \mathcal{F}_{t-1}$ *such that* $|\hat{X}_t| \leq c$ *with probability 1,*

$$\mathbb{P}\left(\exists t \leq T, \exists \{y_k\}_{k=0}^{\infty} \in S : \left|\sum_{k=0}^{t-1} y_k X_k\right| \geq b_t\right) \leq \delta,$$

*where* $b_t \triangleq 8y_t \sqrt{\theta_{t,\delta} \sum_{k=0}^{t-1}(X_k - \hat{X}_k)^2 + c^2\theta_{t,\delta}^2}$ *and* $\theta_{t,\delta} \triangleq \log(60 \log(6t/\delta))$.

**Lemma A.3** (Ivgi et al. (2023a, Lemma C.3)). *Let* $a_{-1}, a_0, \ldots, a_t$ *be a nondecreasing sequence of nonnegative numbers, then the following inequality holds*

$$\sum_{k=0}^{t} \frac{a_k - a_{k-1}}{a_k \log_+^2(a_k/a_{-1})} \leq 1.$$

**Lemma A.4** (Carmon & Hinder (2022, Corollary 1)). *Let* $c > 0$ *and* $X_t$ *be a martingale difference sequence adapted to* $\mathcal{F}_t$ *such that* $|X_t| \leq c$ *with probability 1 for all t. Then, for all* $\delta \in (0, 1)$ , *and* $\hat{X}_t \in \mathcal{F}_{t-1}$ *such that* $|\hat{X}_t| \leq c$ *with probability 1, the following inequality holds*

$$\mathbb{P}\left(\exists t \leq T : \left|\sum_{k=1}^{t} X_k\right| > 4\sqrt{\theta_{t,\delta} \sum_{k=1}^{t}(X_k - \hat{X}_k)^2 + c^2\theta_{t,\delta}^2}\right) \leq \delta.$$

# B. The Proofs for Section 4

We provide detailed proofs for the results under the bounded domain assumption.

## B.1. Proof of Lemma 4.2

*Proof of Lemma 4.2.* For convenience, we omit the subscripts. Recall that the gradient estimator $\mathbf{g}$ is defined as

$$\mathbf{g}(\mathbf{x}, \mu; \mathbf{v}, \boldsymbol{\xi}) = \frac{d}{2\mu}(F(\mathbf{x} + \mu\mathbf{v}; \boldsymbol{\xi}) - F(\mathbf{x} - \mu\mathbf{v}; \boldsymbol{\xi}))\mathbf{v}, \quad \text{where} \quad \mathbf{v} \sim \mathbb{U}(\mathbb{S}^{d-1}).$$

By Assumption 2.6, the function $F(\mathbf{x}; \boldsymbol{\xi})$ is almost surely $L$-Lispchitz in $\mathbf{x}$. Thus, we have

$$\|\mathbf{g}\| = \frac{d}{2\mu}|F(\mathbf{x} + \mu\mathbf{v}; \boldsymbol{\xi}) - F(\mathbf{x} - \mu\mathbf{v}; \boldsymbol{\xi})|\|\mathbf{v}\| \leq Ld\|\mathbf{v}\|^2 = Ld,$$

where the last equality follows from the fact that $\|\mathbf{v}\| = 1$.

Next, using the definition of $\mathbf{g}$ and $\|\mathbf{v}\| = 1$ again, we compute the second moment of $\mathbf{g}$ as

$$\mathbb{E}_{\mathbf{v} \sim \mathbb{U}(\mathbb{S}^{d-1})}[\|\mathbf{g}\|^2] = \frac{d^2}{4\mu^2} \cdot \mathbb{E}_{\mathbf{v} \sim \mathbb{U}(\mathbb{S}^{d-1})}[(F(\mathbf{x} + \mu\mathbf{v}; \boldsymbol{\xi}) - F(\mathbf{x} - \mu\mathbf{v}; \boldsymbol{\xi}))^2].$$

For any $\alpha \in \mathbb{R}$, we can rewrite this as

$$\mathbb{E}_{\mathbf{v} \sim \mathbb{U}(\mathbb{S}^{d-1})}[\|\mathbf{g}\|^2] = \frac{d^2}{4\mu^2} \cdot \mathbb{E}_{\mathbf{v} \sim \mathbb{U}(\mathbb{S}^{d-1})}[((F(\mathbf{x} + \mu\mathbf{v}; \boldsymbol{\xi}) - \alpha) - (F(\mathbf{x} - \mu\mathbf{v}; \boldsymbol{\xi}) - \alpha))^2].$$

Applying the inequality $(a - b)^2 \le 2a^2 + 2b^2$, we obtain

$$\mathbb{E}_{\mathbf{v} \sim \mathbb{U}(\mathbb{S}^{d-1})}[\|\mathbf{g}\|^2] \le \frac{d^2}{2\mu^2} \cdot (\mathbb{E}_{\mathbf{v} \sim \mathbb{U}(\mathbb{S}^{d-1})}[(F(\mathbf{x} + \mu\mathbf{v}; \boldsymbol{\xi}) - \alpha)^2] + \mathbb{E}_{\mathbf{v} \sim \mathbb{U}(\mathbb{S}^{d-1})}[(F(\mathbf{x} - \mu\mathbf{v}; \boldsymbol{\xi}) - \alpha)^2]).$$

Since the distribution $\mathbf{v} \sim \mathbb{U}(\mathbb{S}^{d-1})$ is symmetric about the origin, the two terms on the right-hand side are equal. Thus,

$$\mathbb{E}_{\mathbf{v} \sim \mathbb{U}(\mathbb{S}^{d-1})}[\|\mathbf{g}\|^2] \le \frac{d^2}{\mu^2} \cdot \mathbb{E}_{\mathbf{v} \sim \mathbb{U}(\mathbb{S}^{d-1})}[(F(\mathbf{x} + \mu\mathbf{v}; \boldsymbol{\xi}) - \alpha)^2]. \tag{16}$$

Let $h(\mathbf{v}) \triangleq F(\mathbf{x} + \mu\mathbf{v}; \boldsymbol{\xi})$. Since $F(\mathbf{x}; \boldsymbol{\xi})$ is $L$-Lispchitz in $\mathbf{x}$, it follows that $h(\mathbf{v})$ is $\mu L$-Lipschitz in $\mathbf{v}$. By Lemma A.1, the variance of $h(\mathbf{v})$ is bounded as follows

$$\begin{aligned} \mathbb{E}_{\mathbf{v} \sim \mathbb{U}(\mathbb{S}^{d-1})}[(h(\mathbf{v}) - \mathbb{E}_{\mathbf{v} \sim \mathbb{U}(\mathbb{S}^{d-1})}[h(\mathbf{v})])^2] &= \int_0^\infty \mathbb{P}((h(\mathbf{v}) - \mathbb{E}_{\mathbf{v} \sim \mathbb{U}(\mathbb{S}^{d-1})}[h(\mathbf{v})])^2 > t) \, \mathrm{d}t \\ &= \int_0^\infty \mathbb{P}(|h(\mathbf{v}) - \mathbb{E}_{\mathbf{v} \sim \mathbb{U}(\mathbb{S}^{d-1})}[h(\mathbf{v})]| > \sqrt{t}) \, \mathrm{d}t \\ &\le \int_0^\infty 2 \exp\left(-\frac{cdt}{\mu^2 L^2}\right) \mathrm{d}t = \frac{2\mu^2 L^2}{cd}, \end{aligned}$$

where $c > 0$ is a numerical constant. The first equality follows from the identity (Durrett, 2019, Lemma 2.2.13), which states that if a random variable $Y \ge 0$ almost surely, then

$$\mathbb{E}[Y] = \int_0^\infty \mathbb{P}(Y > y) \, \mathrm{d}y.$$

Setting $\alpha = \mathbb{E}_{\mathbf{v} \sim \mathbb{U}(\mathbb{S}^{d-1})}[h(\mathbf{v})]$, and combining the variance bound above with the inequality (16), we have

$$\mathbb{E}_{\mathbf{v} \sim \mathbb{U}(\mathbb{S}^{d-1})}[\|\mathbf{g}\|^2] \le \frac{d^2}{\mu^2} \cdot \mathbb{E}_{\mathbf{v} \sim \mathbb{U}(\mathbb{S}^{d-1})}[(h(\mathbf{v}) - \mathbb{E}_{\mathbf{v} \sim \mathbb{U}(\mathbb{S}^{d-1})}[h(\mathbf{v})])^2] \le \frac{2}{c} L^2 d.$$

Finally, we apply the law of total expectation to derive $\mathbb{E}[\|\mathbf{g}\|^2] = \mathbb{E}[\mathbb{E}_{\mathbf{v} \sim \mathbb{U}(\mathbb{S}^{d-1})}[\|\mathbf{g}\|^2]] \le 2L^2 d/c.$ $\qquad \square$

## B.2. Proof of Lemma 4.4

*Proof of Lemma 4.4.* We begin by defining the filtration $\mathcal{F}_k \triangleq \sigma(\mathbf{v}_i, \boldsymbol{\xi}_i, 0 \le i \le k)$ for $k \in \mathbb{N}$ and $\mathcal{F}_{-1} \triangleq \{\emptyset, \Omega\}$. Next, we introduce two stochastic processes $(X_k, k \in \mathbb{N})$ and $(\hat{X}_k, k \in \mathbb{N})$ defined as

$$X_k \triangleq \frac{1}{\bar{s}_k} \langle \boldsymbol{\Delta}_k, \mathbf{x}_k - \mathbf{x}_\star \rangle \in \mathcal{F}_k \quad \text{and} \quad \hat{X}_k \triangleq \frac{1}{\bar{s}_k} \langle \nabla f_{\mu_k}(\mathbf{x}_k), \mathbf{x}_k - \mathbf{x}_\star \rangle \in \mathcal{F}_{k-1}, \tag{17}$$

where $\boldsymbol{\Delta}_k = \nabla f_{\mu_k}(\mathbf{x}_k) - \mathbf{g}_k$. Thus, we derive that $(X_k, k \in \mathbb{N})$ is adapted to $(\mathcal{F}_k, k \in \mathbb{N})$ and $(\hat{X}_k, k \in \mathbb{N})$ is predictable with respect to $(\mathcal{F}_k, k \in \mathbb{N})$. Moreover, since $\mathbf{g}_k$ is an unbiased estimator of $\nabla f_{\mu_k}(\mathbf{x}_k)$ conditioned on $\mathcal{F}_{k-1}$, we have

$$\mathbb{E}[X_k \mid \mathcal{F}_{k-1}] = \frac{1}{\bar{s}_k} \cdot \mathbb{E}[\langle \nabla f_{\mu_k}(\mathbf{x}_k) - \mathbf{g}_k, \mathbf{x}_k - \mathbf{x}_\star \rangle \mid \mathcal{F}_{k-1}] = 0,$$

where we use the fact that $\bar{s}_k \in \mathcal{F}_{k-1}$. This implies that $(X_k, \mathcal{F}_k, k \in \mathbb{N})$ forms a martingale difference process.

Next, applying Lemma 4.2, we know that $\|\mathbf{g}_k\| \le Ld$. Since $\mathbf{g}_k$ is an unbiased estimator of $\nabla f_{\mu_k}(\mathbf{x}_k)$, we obtain

$$\|\nabla f_{\mu_k}(\mathbf{x}_k)\| = \|\mathbb{E}_{\mathbf{v}_k, \boldsymbol{\xi}_k}[\mathbf{g}_k]\| \le \mathbb{E}_{\mathbf{v}_k, \boldsymbol{\xi}_k}[\|\mathbf{g}_k\|] \le Ld.$$

It follows that

$$\|\mathbf{\Delta}_k\| \le \|\mathbf{g}_k\| + \|\nabla f_{\mu_k}(\mathbf{x}_k)\| \le 2Ld.$$

From equation (17), we immediately have $|X_k| \le |\mathbf{\Delta}_k| \le 2Ld$ and $|\hat{X}_k| \le |\nabla f_{\mu_k}(\mathbf{x}_k)| \le Ld$.

Now, define the sequence $Y_k := \bar{r}_k \bar{s}_k$ for $k \in \mathbb{N}$, which is non-negative and non-decreasing. Using the concentration inequality for martingale difference sequences from Lemma A.2, and letting $\delta \in (0,1)$ and $c = 2Ld$, we obtain

$$\mathbb{P}\left( \exists t \le T : \left| \sum_{k=0}^{t-1} \bar{r}_k \langle \mathbf{\Delta}_k, \mathbf{x}_k - \mathbf{x}_\star \rangle \right| \ge b_t \right) = \mathbb{P}\left( \exists t \le T : \left| \sum_{k=0}^{t-1} Y_k X_k \right| \right)$$

$$\le \mathbb{P}\left( \exists t \le T, \exists \{y_k\}_{k=0}^{\infty} \in S : \left| \sum_{k=0}^{t-1} y_k X_k \right| \ge b_t \right)$$

$$\le \delta,$$

where $b_t \triangleq 8\bar{r}_{t-1}\bar{s}_{t-1}\sqrt{\theta_{t,\delta}G_{t-1} + 4L^2 d^2\theta_{t,\delta}^2}$, $\theta_{t,\delta} \triangleq \log(60\log(6t/\delta))$ and $S$ is the set of non-negative and non-decreasing sequences. □

## B.3. Proof of Lemma 4.5

*Proof of Lemma 4.5.* Define the partial sum as $S_t \triangleq \sum_{k=1}^{t} 1/\sqrt{k}$ for $t \in \mathbb{N}_+$. An upper bound for $S_t$ can be obtained via the following integral

$$S_t \le 1 + \int_1^t \frac{1}{\sqrt{x}}\,\mathrm{d}x = 2\sqrt{t} - 1 \le 2\sqrt{t}.$$

Using this bound, together with the definition of $\mu_k$ in equation (6), we can bound the noise as follows

$$\sum_{k=0}^{t-1} 2L\bar{r}_k\mu_k \le 2L\bar{r}_{t-1}\sum_{k=0}^{t-1}\mu_k \le 2L\sqrt{d} \cdot \bar{r}_{t-1}^2 S_t = 4L\bar{r}_{t-1}^2\sqrt{dt}.$$

□

## B.4. Proof of Proposition 4.7

*Proof of Proposition 4.7.* Combining Lemma 4.1, 4.4 and 4.5 with equations (9) and (10), we obtain that, with probability at least $1 - \delta$, the upper bound for $f(\bar{\mathbf{x}}_t) - f(\mathbf{x}_\star)$ is given by

$$\frac{(2\bar{s}_t + \bar{r}_t)\sqrt{G_{t-1}} + 8\bar{s}_t\sqrt{\theta_{t,\delta}G_{t-1} + 4L^2 d^2\theta_{t,\delta}^2} + 4\bar{r}_t L\sqrt{dt}}{\sum_{k=0}^{t-1}\bar{r}_k/\bar{r}_t},$$

where we use the fact that $\bar{r}_{t-1} \le \bar{r}_t$ and $\bar{s}_{t-1} \le \bar{s}_t$. Applying the inequality $\sqrt{a^2 + b^2} \le a + b$ to the bound, we have

$$\frac{(2\bar{s}_t + \bar{r}_t)\sqrt{G_{t-1}} + 8\bar{s}_t(\theta_{t,\delta}\sqrt{G_{t-1}} + 2\theta_{t,\delta}Ld) + 4\bar{r}_t L\sqrt{dt}}{\sum_{k=0}^{t-1}\bar{r}_k/\bar{r}_t}.$$

Finally, using the triangle inequality $\bar{s}_t \le \bar{r}_t + s_0$, we obtain the bound

$$16 \cdot \frac{\theta_{t,\delta}(\bar{r}_t + s_0)(\sqrt{G_{t-1}} + Ld + L\sqrt{dt})}{\sum_{k=0}^{t-1}\bar{r}_k/\bar{r}_t}.$$

□

### B.5. Proof of Theorem 4.9

We begin by introducing several useful properties of conditional expectations.

**Lemma B.1.** *Let $(\Omega, \mathcal{F}_0, \mathbb{P})$ be a probability space, and let $X_1, X_2$ be two random variables defined on it. Suppose $\mathcal{F} \subset \mathcal{F}_0$ is a sub-$\sigma$-algebra, and $\mathbb{E}[\cdot \,|\, \mathcal{F}]$ denotes the corresponding conditional expectation. If $X_1 \leq X_2$ on a set $B \in \mathcal{F}$, then $\mathbb{E}[X_1 \,|\, \mathcal{F}] \leq \mathbb{E}[X_2 \,|\, \mathcal{F}]$ almost surely on $B$.*

*Proof of Lemma B.1.* We follow the proof strategy of Durrett (2019, Theorem 4.1.2). For any $\epsilon > 0$, define the event $A = \{\omega \in \Omega : \mathbb{E}[X_1 \,|\, \mathcal{F}] - \mathbb{E}[X_2 \,|\, \mathcal{F}] \geq \epsilon\}$, which satisfies $A \in \mathcal{F}$ since both conditional expectations are $\mathcal{F}$-measurable. Because $B \in \mathcal{F}$, their intersection $A \cap B \in \mathcal{F}$. Then, by the definition of conditional expectation, we have

$$\int_{A \cap B} \mathbb{E}[X_1 \,|\, \mathcal{F}] - \mathbb{E}[X_2 \,|\, \mathcal{F}] \, d\mathbb{P} = \int_{A \cap B} X_1 - X_2 \, d\mathbb{P} \leq 0,$$

where the inequality follows from the assumption $X_1 \leq X_2$ on $B$. On the other hand, by the definition of $A$, we have

$$\int_{A \cap B} \mathbb{E}[X_1 \,|\, \mathcal{F}] - \mathbb{E}[X_2 \,|\, \mathcal{F}] \, d\mathbb{P} \geq \int_{A \cap B} \epsilon \, d\mathbb{P} = \epsilon \cdot \mathbb{P}(A \cap B).$$

Combining the two inequality, we have $\mathbb{P}(A \cap B) = 0$, which implies that

$$\mathbb{P}(\omega \in B : \mathbb{E}[X_1 \,|\, \mathcal{F}] - \mathbb{E}[X_2 \,|\, \mathcal{F}] \geq \epsilon) = 0.$$

Since this holds for all $\epsilon > 0$, it follows that $\mathbb{E}[X_1 \,|\, \mathcal{F}] \leq \mathbb{E}[X_2 \,|\, \mathcal{F}]$ almost surely on $B$. $\qquad\square$

**Lemma B.2** (Durrett (2019, Theorem 4.1.13)). *If $\mathcal{F}_0 \subset \mathcal{F}$, then $\mathbb{E}[X \,|\, \mathcal{F}_0] = \mathbb{E}[\mathbb{E}[X \,|\, \mathcal{F}] \,|\, \mathcal{F}_0]$.*

**Lemma B.3** (Durrett (2019, Theorem 4.1.10)). *Let $\phi$ be a convex function and $X$ be a random variable such that $\mathbb{E}|X| < \infty$ and $\mathbb{E}|\phi(X)| < \infty$. Then,*

$$\phi(\mathbb{E}[X \,|\, \mathcal{F}]) \leq \mathbb{E}[\phi(X) \,|\, \mathcal{F}].$$

We are now ready to prove Theorem 4.9.

*Proof of Theorem 4.9.* Recall the event

$$\Omega_\delta \triangleq \left\{ \omega \in \Omega_0 : \forall t \leq T, \left| \sum_{k=0}^{t-1} \bar{r}_k \langle \mathbf{\Delta}_k, \mathbf{x}_k - \mathbf{x}_\star \rangle \right| < b_t \right\} \in \mathcal{F}_0,$$

which satisfies $\mathbb{P}(\Omega_\delta) \geq 1 - \delta$ by Lemma 4.4. From Proposition 4.7, we know that for any $\omega \in \Omega_\delta$ and all $t \leq T$, the following inequalty holds

$$f(\bar{\mathbf{x}}_t) - f(\mathbf{x}_\star) \leq 16 \cdot \frac{\theta_{t,\delta}(\bar{r}_t + s_0)(\sqrt{G_{t-1}} + Ld + L\sqrt{dt})}{\sum_{k=0}^{t-1} \bar{r}_k / \bar{r}_t}.$$

Combining this with equation (11), we obtain that for all $\omega \in \Omega_\delta$,

$$f(\bar{\mathbf{x}}_{\tau_T}) - f(\mathbf{x}_\star) \leq c_0 \cdot \frac{\theta_{\tau_T,\delta}(\bar{r}_{\tau_T} + s_0)(\sqrt{G_{\tau_T-1}} + Ld + L\sqrt{d\tau_T})}{T} \log_+\left(\frac{\bar{r}_{\tau_T}}{r_\epsilon}\right),$$

where $c_0$ is a constant and $\tau_T = \arg\max_{t \leq T} \sum_{k=0}^{t-1} \bar{r}_k / \bar{r}_t$. Since $\theta_{t,\delta}$, $\bar{r}_t$, and $G_t$ are non-decreasing in $t$ and $\tau_T \leq T$, we can simplify the bound

$$f(\bar{\mathbf{x}}_{\tau_T}) - f(\mathbf{x}_\star) \leq c_0 \cdot \frac{\theta_{T,\delta}(\bar{r}_T + s_0)(\sqrt{G_{T-1}} + Ld + L\sqrt{dT})}{T} \log_+\left(\frac{\bar{r}_T}{r_\epsilon}\right).$$

Noting that the diameter is $D_{\mathcal{X}}$ and $r_\epsilon \leq D_{\mathcal{X}}$, we have $\bar{r}_T \leq D_{\mathcal{X}}$. Substituting this into the above bound yields

$$f(\bar{\mathbf{x}}_{\tau_T}) - f(\mathbf{x}_\star) \leq 2c_0 \cdot \frac{\theta_{T,\delta} D_{\mathcal{X}}(\sqrt{G_{T-1}} + Ld + L\sqrt{dT})}{T} \log_+ \left( \frac{D_{\mathcal{X}}}{r_\epsilon} \right).$$

Recall that $\mathcal{F}_\delta \triangleq \{A : A \subset \Omega_\delta\} \cap \mathcal{F}_0$ is a sigma field satisfying $\mathcal{F}_\delta \subset \mathcal{F}_0$. Moreover, we have $\Omega_\delta \in \mathcal{F}_\delta$. Applying Lemma B.1, we obtain that for any $\omega \in \Omega_\delta$,

$$\mathbb{E}[f(\bar{\mathbf{x}}_{\tau_T}) - f(\mathbf{x}_\star) \,|\, \mathcal{F}_\delta] \leq 2c_0 \cdot \frac{\theta_{T,\delta} D_{\mathcal{X}}(\mathbb{E}[\sqrt{G_{T-1}} \,|\, \mathcal{F}_\delta] + Ld + L\sqrt{dT})}{T} \log_+ \left( \frac{D_{\mathcal{X}}}{r_\epsilon} \right). \tag{18}$$

Next, we bound the conditional expectation of $G_{T-1}$ using Lemma 4.2 and Lemma B.2 as follows

$$\mathbb{E}[G_{T-1} \,|\, \mathcal{F}_\delta] = \mathbb{E}[\mathbb{E}[G_{T-1}] \,|\, \mathcal{F}_\delta] = \sum_{k=0}^{T-1} \mathbb{E}[\mathbb{E}[\|\mathbf{g}_k\|^2] \,|\, \mathcal{F}_\delta] \leq cL^2 dT.$$

Since the square root function is concave, applying Jensen's inequality (Lemma B.3) gives

$$\mathbb{E}[\sqrt{G_{T-1}} \,|\, \mathcal{F}_\delta] \leq \sqrt{\mathbb{E}[G_{T-1} \,|\, \mathcal{F}_\delta]} \leq L\sqrt{cdT}.$$

Substituting this back into equation (18), we conclude that , for any $\omega \in \Omega_\delta$, the following inequality holds

$$\mathbb{E}[f(\bar{\mathbf{x}}_{\tau_T}) - f(\mathbf{x}_\star) \,|\, \mathcal{F}_\delta] \leq c_1 \left( \frac{d}{T} + \frac{\sqrt{d}}{\sqrt{T}} \right) \theta_{T,\delta} L D_{\mathcal{X}} \log_+ \left( \frac{D_{\mathcal{X}}}{r_\epsilon} \right), \tag{19}$$

where $c_1$ is a constant. This implies that the upper bound (19) holds with probability at least $1 - \delta$. $\qquad \square$

## C. The Proofs for Section 5

We provide detailed proofs for the results without assuming a bounded domain.

### C.1. Proof of Proposition 5.3

For simplicity, we define the following stopping time

$$\zeta \triangleq \min\{t \in \mathbb{N} \,|\, \bar{r}_t > 3s_0\}.$$

Using this stopping time, we define a modified step size

$$\tilde{\eta}_t \triangleq \eta_t \cdot \mathbb{I}(t < \zeta),$$

where the indicator function $\mathbb{I}(t < \zeta)$ equals 1 if $t < \zeta$, and 0 otherwise.

Before proving the proposition, we first present and prove several supporting lemmas.

**Lemma C.1.** *Let $T \in \mathbb{N}+$. For any $t \leq T$, the following inequality holds*

$$\sum_{k=0}^{t} \tilde{\eta}_k^2 \|\mathbf{g}_k\|^2 \leq \frac{s_0^2}{2}.$$

*Proof of Lemma C.1.* By the definition of $\tilde{\eta}_k$ and using the identity $\|\mathbf{g}_k\|^2 = G_k - G_{k-1}$, we can bound the sum as follows

$$\sum_{k=0}^{t} \tilde{\eta}_k^2 \|\mathbf{g}_k\|^2 \leq \sum_{k=0}^{\zeta-1} \eta_k^2 \|\mathbf{g}_k\|^2 = \sum_{k=0}^{\zeta-1} \frac{\bar{r}_k^2}{G_k'} \cdot \|\mathbf{g}_k\|^2 = \sum_{k=0}^{\zeta-1} \frac{\bar{r}_k^2 (G_k - G_{k-1})}{G_k'} \leq \bar{r}_{\zeta-1}^2 \sum_{k=0}^{\zeta-1} \frac{G_k - G_{k-1}}{G_k'}, \tag{20}$$

where we set $G_{-1} = 0$. We now use a lower bound for $G_k'$

$$G_k' \geq 8^4 \theta_{T,\delta}(G_{k-1} + 2d^2 \bar{L}^2) \log_+^2 \left( \frac{(k+1)d^2 \bar{L}^2 + d^2 \bar{L}^2}{d^2 \bar{L}^2} \right) \geq 8^4 \theta_{T,\delta}(G_k + d^2 \bar{L}^2) \log_+^2 \left( \frac{G_k + d^2 \bar{L}^2}{d^2 \bar{L}^2} \right),$$

where the last inequality follows from $\|\mathbf{g}_k\| \leq Ld$ ( See Lemma 4.2). Substituting this bound into (20), we obtain

$$\sum_{k=0}^{t} \tilde{\eta}_k^2 \|\mathbf{g}_k\|^2 \leq \frac{\bar{r}_{\zeta-1}^2}{8^4 \theta_{T,\delta}} \cdot \sum_{k=0}^{\zeta-1} \frac{G_k - G_{k-1}}{(G_k + d^2 \bar{L}^2) \log_+^2 \left( \frac{G_k + d^2 \bar{L}^2}{d^2 \bar{L}^2} \right)} \leq \frac{\bar{r}_{\zeta-1}^2}{8^4 \theta_{T,\delta}} \leq \frac{9 s_0^2}{8^4 \theta_{T,\delta}} \leq \frac{s_0^2}{2}. \tag{21}$$

The second inequality holds by applying Lemma A.3 with $a_k = G_k + d^2 \bar{L}^2$. $\qquad\square$

**Lemma C.2.** *For any* $\delta \in (0, 1)$*, the following inequality holds*

$$\mathbb{P}\left( \exists t \leq T : \left| \sum_{k=0}^{t-1} \tilde{\eta}_k \langle \boldsymbol{\Delta}_k, \mathbf{x}_k - \mathbf{x}_\star \rangle \right| > s_0^2 \right) \leq \delta,$$

*Proof of Lemma C.2.* We consider the filtration $\mathcal{F}_k = \sigma(\mathbf{v}_i, \boldsymbol{\xi}_i, 0 \leq i \leq k)$ for $k \in \mathbb{N}$ and $\mathcal{F}_{-1} = \{\emptyset, \Omega_0\}$ as defined in Appendix B.2. Note that $\tilde{\eta}_k \in \mathcal{F}_{k-1}$. Define the stochastic processes $(Z_k, k \in \mathbb{N})$ and $(\hat{Z}_k, k \in \mathbb{N})$ as

$$Z_k = \tilde{\eta}_k \langle \boldsymbol{\Delta}_k, \mathbf{x}_k - \mathbf{x}_\star \rangle \in \mathcal{F}_k \quad \text{and} \quad \hat{Z}_k = \tilde{\eta}_k \langle \nabla f_{\mu_k}(\mathbf{x}_k), \mathbf{x}_k - \mathbf{x}_\star \rangle \in \mathcal{F}_{k-1},$$

where $\boldsymbol{\Delta}_k = \nabla f_{\mu_k}(\mathbf{x}_k) - \mathbf{g}_k$. By construction, we have

$$\mathbb{E}[Z_k \,|\, \mathcal{F}_{k-1}] = \tilde{\eta}_k \cdot \mathbb{E}[\langle \nabla f_{\mu_k}(\mathbf{x}_k) - \mathbf{g}_k, \mathbf{x}_k - \mathbf{x}_\star \rangle \,|\, \mathcal{F}_{k-1}] = 0, \quad \text{where} \quad k \in \mathbb{N}.$$

Thus, $(Z_k, \mathcal{F}_k, k \in \mathbb{N})$ is a martingale difference process.

We now bound $|Z_k|$. Using the fact that $\bar{s}_t \leq \bar{r}_t + s_0$, we obtain

$$|Z_k| \leq \tilde{\eta}_k s_k \|\boldsymbol{\Delta}_k\| \leq \frac{\bar{r}_{\zeta-1} \bar{s}_{\zeta-1} \|\boldsymbol{\Delta}_k\|}{\sqrt{G_k'}} \leq \frac{12 s_0^2 \|\boldsymbol{\Delta}_k\|}{\sqrt{G_k'}}.$$

From Appendix B.2, we have $\|\boldsymbol{\Delta}_k\| \leq 2Ld$. Moreover, we have $G_k' \geq 16 \cdot 8^4 \theta_{T,\delta}^2 d^2 \bar{L}^2$. Therefore, we conclude

$$|Z_k| \leq \frac{6 s_0^2}{8^2 \theta_{T,\delta}}.$$

The same upper bound also applies to $|\hat{Z}_k|$. Now apply Lemma A.4 with $c = 6 s_0^2 / (8^2 \theta_{T,\delta})$. This gives

$$\mathbb{P}\left( \exists t \leq T : \left| \sum_{k=0}^{t-1} Z_k \right| > 4 \sqrt{\theta_{t,\delta} \sum_{k=0}^{t-1} (Z_k - \hat{Z}_k)^2 + c^2 \theta_{t,\delta}^2} \right) \leq \delta. \tag{22}$$

The upper bound for $\sum_{k=0}^{t-1} (Z_k - \hat{Z}_k)^2$ is given by

$$\sum_{k=0}^{t-1} (Z_k - \hat{Z}_k)^2 = \sum_{k=0}^{t-1} \tilde{\eta}_k^2 (\langle \mathbf{g}_k, \mathbf{x}_k - \mathbf{x}_\star \rangle)^2 \leq \sum_{k=0}^{t-1} \tilde{\eta}_k^2 s_k^2 \|\mathbf{g}_k\|^2 \leq \bar{s}_{\zeta-1}^2 \sum_{k=0}^{t-1} \tilde{\eta}_k^2 \|\mathbf{g}_k\|^2 \leq (4s_0)^2 \sum_{k=0}^{t-1} \tilde{\eta}_k^2 \|\mathbf{g}_k\|^2 \leq \frac{12^2 s_0^4}{8^4 \theta_{T,\delta}}, \tag{23}$$

where the third inequality follows from $\bar{s}_k \leq \bar{r}_k + s_0$ and the last follows from (21). Substituting (23) into (22), we get

$$4 \sqrt{\theta_{t,\delta} \sum_{k=0}^{t-1} (Z_k - \hat{Z}_k)^2 + c^2 \theta_{t,\delta}^2} \leq 4 \sqrt{\theta_{t,\delta} \frac{12^2 s_0^4}{8^4 \theta_{T,\delta}} + \theta_{t,\delta}^2 \frac{6^2 s_0^4}{8^4 \theta_{T,\delta}^2}} \leq 4 \sqrt{\frac{12^2 s_0^4}{8^4} + \frac{6^2 s_0^4}{8^4}} \leq s_0^2.$$

Thus, we have

$$\mathbb{P}\left( \exists t \leq T : \left| \sum_{k=0}^{t-1} \tilde{\eta}_k \langle \boldsymbol{\Delta}_k, \mathbf{x}_k - \mathbf{x}_\star \rangle \right| > s_0^2 \right) \leq \delta.$$

$\qquad\square$

**Lemma C.3.** *Let $T \in \mathbb{N}+$. For any $t \leq T$, the following inequality holds*

$$\sum_{k=0}^{t} \tilde{\eta}_k \langle \nabla f_{\mu_k}(\mathbf{x}_k), \mathbf{x}_\star - \mathbf{x}_k \rangle \leq \frac{s_0^2}{4}.$$

*Proof of Lemma C.3.* By Lemma 2.8, we have

$$\langle \nabla f_{\mu_k}(\mathbf{x}_k), \mathbf{x}_\star - \mathbf{x}_k \rangle \leq f_{\mu_k}(\mathbf{x}_\star) - f_{\mu_k}(\mathbf{x}_k) \leq f(\mathbf{x}_\star) - f(\mathbf{x}_k) + 2L\mu_k \leq 2L\mu_k,$$

where the last inequality follows from the fact that $f(\mathbf{x}_\star) = \inf_{\mathbf{x} \in \mathcal{X}} f(\mathbf{x})$. Thus, the summation can be bounded as follows

$$\sum_{k=0}^{t} \tilde{\eta}_k \langle \nabla f_{\mu_k}(\mathbf{x}_k), \mathbf{x}_\star - \mathbf{x}_k \rangle = \sum_{k=0}^{\min(\zeta-1,t)} \eta_k \langle \nabla f_{\mu_k}(\mathbf{x}_k), \mathbf{x}_\star - \mathbf{x}_k \rangle \leq 2L \sum_{k=0}^{\min(\zeta-1,t)} \eta_k \mu_k.$$

Given that $\mu_k = d\bar{r}_k/(k+1)^2$ and $G'_k \geq 16 \cdot 8^4 d^2 \bar{L}^2$, it follows that

$$\sum_{k=0}^{t} \tilde{\eta}_k \langle \nabla f_{\mu_k}(\mathbf{x}_k), \mathbf{x}_\star - \mathbf{x}_k \rangle \leq \frac{2L}{4 \cdot 8^2 \bar{L}} \sum_{k=0}^{\min(\zeta-1,t)} \frac{\bar{r}_k^2}{(k+1)^2} \leq \frac{\bar{r}_{\zeta-1}^2}{2 \cdot 8^2} \sum_{k=0}^{\min(\zeta-1,t)} \frac{1}{(k+1)^2} \leq \frac{3\pi^2 s_0^2}{4 \cdot 8^2} \leq \frac{s_0^2}{4},$$

where the third inequality uses the fact that $\sum_{k=1}^{\infty} 1/k^2 = \pi^2/6$ and $\bar{r}_{\zeta-1} \leq 3s_0$. $\qquad\square$

Based on Lemmas C.1, C.2, and C.3, we now establish Proposition 5.3.

*Proof of Proposition 5.3.* Fix any $\delta > 0$, and define the event

$$\tilde{\Omega}_\delta \triangleq \left\{ \omega \in \Omega_0 : \forall t \leq T, \left| \sum_{k=0}^{t-1} \tilde{\eta}_k \langle \mathbf{\Delta}_k, \mathbf{x}_k - \mathbf{x}_\star \rangle \right| \leq s_0^2 \right\}.$$

By Lemma C.2, it holds that $\mathbb{P}(\tilde{\Omega}_\delta) \geq 1 - \delta$.

We now proceed by induction on $t$ to show that $\bar{r}_t \leq 3s_0$ for all $t \leq T$ and any $\omega \in \tilde{\Omega}_\delta$. For the base case, we have $\bar{r}_0 = r_\epsilon \leq 3s_0$. For the induction step, we assume $\bar{r}_{t-1} \leq 3s_0$, which implies that $\zeta > t - 1$. From equation (14), we have

$$\begin{aligned}
s_t^2 - s_0^2 &\leq \sum_{k=0}^{t-1} \eta_k^2 \|\mathbf{g}_k\|^2 + 2\sum_{k=0}^{t-1} \eta_k \langle \mathbf{\Delta}_k, \mathbf{x}_k - \mathbf{x}_\star \rangle + 2\sum_{k=0}^{t-1} \eta_k \langle \nabla f_{\mu_k}(\mathbf{x}_k), \mathbf{x}_\star - \mathbf{x}_k \rangle \\
&= \sum_{k=0}^{t-1} \tilde{\eta}_k^2 \|\mathbf{g}_k\|^2 + 2\sum_{k=0}^{t-1} \tilde{\eta}_k \langle \mathbf{\Delta}_k, \mathbf{x}_k - \mathbf{x}_\star \rangle + 2\sum_{k=0}^{t-1} \tilde{\eta}_k \langle \nabla f_{\mu_k}(\mathbf{x}_k), \mathbf{x}_\star - \mathbf{x}_k \rangle,
\end{aligned}$$

where the equality holds since $\zeta > t - 1$. Now, applying Lemmas C.1, C.2, and C.3, we obtain for any $\omega \in \tilde{\Omega}_\delta$

$$s_t^2 - s_0^2 \leq \frac{1}{2}s_0^2 + 2s_0^2 + 2 \cdot \frac{1}{4}s_0^2 = 3s_0^2,$$

which implies that $s_t \leq 2s_0$. Hence, we have $r_t \leq s_t + s_0 = 3s_0$. By induction, it follows that $\bar{r}_t = \max(\bar{r}_{t-1}, r_t) \leq 3s_0$ for all $t \leq T$ for any $\omega \in \tilde{\Omega}_\delta$. Equivalently, we have $\mathbb{P}(\bar{r}_T > 3s_0) \leq \delta$. $\qquad\square$

## C.2. Proof of Proposition 5.4

*Proof of Proposition 5.4.* Since $G'_t \geq G_t$, we can apply the result of Ivgi et al. (2023a, Lemma 3.4), replacing $G_t$ with $G'_t$, which ensures that Lemma 4.1 still holds in our setting. Moreover, Lemma 4.4 remains valid. For Lemma 4.5, recall that $\mu_t = d\bar{r}_t/(t+1)^2$ as given in equation (13). The noise from $\mu$ can be bounded as

$$\sum_{k=0}^{t-1} 2L\bar{r}_k \mu_k = \sum_{k=0}^{t-1} \frac{2Ld\bar{r}_k^2}{(k+1)^2} \leq 2Ld\bar{r}_{t-1}^2 \sum_{k=0}^{t-1} \frac{1}{(k+1)^2} \leq 4Ld\bar{r}_{t-1}^2,$$

where the last inequality uses the fact that $\sum_{k=1}^{\infty} 1/k^2 = \pi^2/6$.

Combining the modified lemmas and using equations (9) and (10), we obtain that, with probability at least $1 - \delta$, the following upper bound on the optimality gap holds

$$\frac{(2\bar{s}_t + \bar{r}_t)\sqrt{G'_{t-1}} + 8\bar{s}_t\sqrt{\theta_{t,\delta}G_{t-1} + 4L^2d^2\theta_{t,\delta}^2} + 4Ld\bar{r}_t}{\sum_{k=0}^{t-1}\bar{r}_k/\bar{r}_t},$$

where we use the fact that $\bar{r}_{t-1} \le \bar{r}_t$ and $\bar{s}_{t-1} \le \bar{s}_t$. Applying the inequality $\sqrt{a^2 + b^2} \le a + b$, the gap simplifies to

$$\frac{(2\bar{s}_t + \bar{r}_t)\sqrt{G'_{t-1}} + 8\bar{s}_t(\theta_{t,\delta}\sqrt{G_{t-1}} + 2\theta_{t,\delta}Ld) + 4Ld\bar{r}_t}{\sum_{k=0}^{t-1}\bar{r}_k/\bar{r}_t}.$$

Finally, using the fact that $G'_t \ge G_t$ and the triangle inequality $\bar{s}_t \le \bar{r}_t + s_0$, the gap becomes

$$20 \cdot \frac{\theta_{t,\delta}(\bar{r}_t + s_0)(\sqrt{G'_{t-1}} + Ld)}{\sum_{k=0}^{t-1}\bar{r}_k/\bar{r}_t}.$$

$\square$

## C.3. Proof of Theorem 5.5

We begin by establishing the existence and uniqueness of the conditional expectation $\mathbb{E}[f(\bar{\mathbf{x}}_{\tau_T}) - f(\mathbf{x}_\star) \,|\, \tilde{\mathcal{F}}_\delta]$. According to Durrett (2019, Chapter 4.1), it suffices to verify that $\tilde{\mathcal{F}}_\delta \subset \tilde{\mathcal{F}}_0$ and $\mathbb{E}|f(\bar{\mathbf{x}}_{\tau_T}) - f(\mathbf{x}_\star)| < \infty$. By definition, we recall that $\tilde{\mathcal{F}}_\delta = \{A : A \subset \hat{\Omega}_\delta \cap \hat{\Omega}_\delta\} \cap \tilde{\mathcal{F}}_0$, which implies that $\tilde{\mathcal{F}}_\delta \subset \tilde{\mathcal{F}}_0$. To verify the integrability condition, we start from inequality (14), which states

$$s_t^2 - s_0^2 \le \sum_{k=0}^{t-1}\eta_k^2\|\mathbf{g}_k\|^2 + 2\sum_{k=0}^{t-1}\eta_k\langle\boldsymbol{\Delta}_k, \mathbf{x}_k - \mathbf{x}_\star\rangle + 2\sum_{k=0}^{t-1}\eta_k\langle\nabla f_{\mu_k}(\mathbf{x}_k), \mathbf{x}_\star - \mathbf{x}_k\rangle,$$

for $t = 1, 2, \ldots, T$. Applying the upper bounds $\|\mathbf{g}_k\| \le Ld$ (Lemma 4.2), $\|\nabla f_{\mu_k}(\mathbf{x}_k)\| \le Ld$ and $\|\boldsymbol{\Delta}_k\| \le 2Ld$ (Appendix B.2), we obtain

$$s_t^2 - s_0^2 \le \sum_{k=0}^{t-1}\eta_k^2\|\mathbf{g}_k\|^2 + 2\sum_{k=0}^{t-1}\eta_k\|\boldsymbol{\Delta}_k\|\|\mathbf{x}_k - \mathbf{x}_\star\| + 2\sum_{k=0}^{t-1}\eta_k\|\nabla f_{\mu_k}(\mathbf{x}_k)\|\|\mathbf{x}_\star - \mathbf{x}_k\|$$

$$\le L^2d^2\sum_{k=0}^{t-1}\eta_k^2 + 6Ld\bar{s}_{t-1}\sum_{k=0}^{t-1}\eta_k.$$

Since $\eta_k = \bar{r}_k/\sqrt{G'_k}$ with $G'_k \ge d^2L^2$, it follows that

$$s_t^2 - s_0^2 \le T(\bar{r}_{t-1}^2 + 6\bar{r}_{t-1}\bar{s}_{t-1}).$$

Given that $\bar{r}_0 = r_\epsilon \le 3s_0 < \infty$, it follows by induction that $\bar{s}_t < \infty$ and $\bar{r}_t < \infty$ for all $t \le T$. Finally, by the Lipschitz continuity of $f(\cdot)$, we obtain

$$|f(\bar{\mathbf{x}}_{\tau_T}) - f(\mathbf{x}_\star)| \le L\|\bar{\mathbf{x}}_{\tau_T} - \mathbf{x}_\star\| \le L\bar{s}_T < \infty.$$

Hence, $\mathbb{E}|f(\bar{\mathbf{x}}_{\tau_T}) - f(\mathbf{x}_\star)| < \infty$. Therefore, we conclude that the conditional expectation $\mathbb{E}[f(\bar{\mathbf{x}}_{\tau_T}) - f(\mathbf{x}_\star) \,|\, \tilde{\mathcal{F}}_\delta]$ exists and is unique.

Now, we provide the proof of Theorem 5.5.

*Proof of Theorem 5.5.* Recall the definitions

$$\tilde{\Omega}_\delta = \left\{\omega \in \Omega_0 : \forall t \le T, \left|\sum_{k=0}^{t-1}\tilde{\eta}_k\langle\boldsymbol{\Delta}_k, \mathbf{x}_k - \mathbf{x}_\star\rangle\right| \le s_0^2\right\},$$

$$\hat{\Omega}_\delta = \left\{ \omega \in \Omega_0 : \forall t \leq T, \left| \sum_{k=0}^{t-1} \bar{r}_k \langle \mathbf{\Delta}_k, \mathbf{x}_k - \mathbf{x}_\star \rangle \right| < b_t \right\},$$

with $\tilde{\mathbb{P}}(\tilde{\Omega}_\delta) \geq 1 - \delta$ and $\tilde{\mathbb{P}}(\hat{\Omega}_\delta) \geq 1 - \delta$. By Proposition 5.4, for any $\omega \in \tilde{\Omega}_\delta \cap \hat{\Omega}_\delta$, we have

$$f(\bar{\mathbf{x}}_t) - f(\mathbf{x}_\star) \leq 20 \cdot \frac{\theta_{t,\delta}(\bar{r}_t + s_0)(\sqrt{G'_{t-1}} + Ld)}{\sum_{k=0}^{t-1} \bar{r}_k / \bar{r}_t}, \quad t \leq T.$$

Combining this with equation (11), we obtain that for any $\omega \in \tilde{\Omega}_\delta \cap \hat{\Omega}_\delta$, the following inequality holds

$$f(\bar{\mathbf{x}}_{\tau_T}) - f(\mathbf{x}_\star) \leq c_0 \cdot \frac{\theta_{\tau_T,\delta}(\bar{r}_{\tau_T} + s_0)(\sqrt{G'_{\tau_T-1}} + Ld)}{T} \log_+\left( \frac{\bar{r}_{\tau_T}}{r_\epsilon} \right),$$

where $c_0$ is a constant and $\tau_T = \arg\max_{t \leq T} \sum_{k=0}^{t-1} \bar{r}_k / \bar{r}_t$. Since $\theta_{t,\delta}$, $\bar{r}_t$ and $G'_t$ are non-deceasing in $t$ and $\tau_T \leq T$, we derive

$$f(\bar{\mathbf{x}}_{\tau_T}) - f(\mathbf{x}_\star) \leq c_0 \cdot \frac{\theta_{T,\delta}(\bar{r}_T + s_0)\left(\sqrt{G'_{T-1}} + Ld\right)}{T} \log_+\left( \frac{\bar{r}_T}{r_\epsilon} \right).$$

Moreover, from Proposition 5.3, we know that $\bar{r}_T \leq 3s_0$ for any $\omega \in \tilde{\Omega}_\delta \cap \hat{\Omega}_\delta$. Substituting this yields

$$f(\bar{\mathbf{x}}_{\tau_T}) - f(\mathbf{x}_\star) \leq 4c_0 \cdot \frac{\theta_{T,\delta} s_0 \left(\sqrt{G'_{T-1}} + Ld\right)}{T} \log_+\left( \frac{3s_0}{r_\epsilon} \right).$$

Recall that $\tilde{\mathcal{F}}_\delta = \{A : A \subset \tilde{\Omega}_\delta \cap \hat{\Omega}_\delta\} \cap \tilde{\mathcal{F}}_0$ is a sigma field satisfying $\tilde{\mathcal{F}}_\delta \subset \tilde{\mathcal{F}}_0$ and $\tilde{\Omega}_\delta \cap \hat{\Omega}_\delta \in \tilde{\mathcal{F}}_\delta$. Then, applying Lemma B.1, for any $\omega \in \tilde{\Omega}_\delta \cap \hat{\Omega}_\delta$, we have

$$\mathbb{E}[f(\bar{\mathbf{x}}_{\tau_T}) - f(\mathbf{x}_\star) \,|\, \tilde{\mathcal{F}}_\delta] \leq 4c_0 \cdot \frac{\theta_{T,\delta} s_0 \left(\mathbb{E}\left[\sqrt{G'_{T-1}} \,\middle|\, \tilde{\mathcal{F}}_\delta\right] + Ld\right)}{T} \log_+\left( \frac{3s_0}{r_\epsilon} \right). \tag{24}$$

Using Lemma 4.2 and Lemma B.2, we can bound the conditional expectation of $G'_{T-1}$ as follows

$$\mathbb{E}[G'_{T-1} \,|\, \tilde{\mathcal{F}}_\delta] = \mathbb{E}[\mathbb{E}[G'_{T-1}] \,|\, \tilde{\mathcal{F}}_\delta] = 8^4 \theta_{T,\delta} \log_+^2(T+1) \left( \sum_{k=0}^{T-1} \mathbb{E}[\mathbb{E}[\|\mathbf{g}_k\|^2] \,|\, \tilde{\mathcal{F}}_\delta] + 16\theta_{T,\delta} d^2 \bar{L}^2 \right)$$

$$\leq 8^4 \theta_{T,\delta}^2 \log_+^2(T+1)(cL^2 dT + 16d^2\bar{L}^2).$$

Since the square root function is concave, applying Jensen's inequality in Lemma B.3 gives

$$\mathbb{E}[\sqrt{G'_{T-1}} \,|\, \tilde{\mathcal{F}}_\delta] \leq \sqrt{\mathbb{E}[G'_{T-1} \,|\, \tilde{\mathcal{F}}_\delta]} \leq 8^2 \theta_{T,\delta} \log_+(T+1)\sqrt{cL^2 dT + 16d^2\bar{L}^2} \leq 8^2 \theta_{T,\delta} \log_+(T+1)(\sqrt{c}L\sqrt{dT} + 4d\bar{L}).$$

Substituting this back into equation (24), for any $\omega \in \tilde{\Omega}_\delta \cap \hat{\Omega}_\delta$, we have

$$\mathbb{E}[f(\bar{\mathbf{x}}_{\tau_T}) - f(\mathbf{x}_\star) \,|\, \tilde{\mathcal{F}}_\delta] \leq c_1 \left( \frac{d}{T} \cdot (L + \bar{L}) + \frac{\sqrt{d}}{\sqrt{T}} \cdot L \right) \alpha_{T,\delta} s_0 \log_+\left( \frac{s_0}{r_\epsilon} \right), \tag{25}$$

where $c_1$ is a constant and $\alpha_{T,\delta} = \theta_{T,\delta} \log_+(T+1)$. Moreover, we have

$$\tilde{\mathbb{P}}(\tilde{\Omega}_\delta \cap \hat{\Omega}_\delta) \geq \tilde{\mathbb{P}}(\tilde{\Omega}_\delta) + \tilde{\mathbb{P}}(\hat{\Omega}_\delta) - 1 \geq 1 - 2\delta.$$

$\square$

## C.4. Proof of Theorem 5.6

*Proof.* For a given $T \geq 2$, let $\boldsymbol{\xi} \sim \Xi$, where $\Xi$ is a Bernoulli distribution defined by

$$\mathbb{P}(\boldsymbol{\xi} = 0) = 1 - \frac{1}{T} \quad \text{and} \quad \mathbb{P}(\boldsymbol{\xi} = 1) = \frac{1}{T}.$$

We first define the function $f_1 : \mathbb{R}^d \to \mathbb{R}$ as

$$f_1(\mathbf{x}) = L\|\mathbf{x}\|_1,$$

where $\|\cdot\|_1$ is the $\ell_1$-norm. The SZO for function $f_1$ is constructed such that for any $\mathbf{x} \in \mathbb{R}^d$ and $\mathbf{y} \in \mathbb{R}^d$, the oracle returns evaluations $F_1(\mathbf{x}; \boldsymbol{\xi})$ and $F_1(\mathbf{y}; \boldsymbol{\xi})$ satisfying $F_1(\mathbf{x}; \boldsymbol{\xi}) = L\|\mathbf{x}\|_1$ and $F_2(\mathbf{y}; \boldsymbol{\xi}) = L\|\mathbf{y}\|_1$ for all $\boldsymbol{\xi}$ drawn from $\Xi$. This oracle clearly satisfies Assumption 2.7. Moreover, the function $F_1(\cdot; \boldsymbol{\xi})$ is convex and Lipschitz continuous with $L_1 = L$ for all $\boldsymbol{\xi}$.

Next, we define another function $f_2 : \mathbb{R}^d \to \mathbb{R}$ as

$$f_2(\mathbf{x}) = L\|\mathbf{x} - \mathbf{u}\|_1,$$

where $\mathbf{u} = (1 - 1/T)\mathbf{1}_d \in \mathbb{R}^d$. The SZO for function $f_2$ is constructed such that for any $\mathbf{x} \in \mathbb{R}^d$ and $\mathbf{y} \in \mathbb{R}^d$, it returns the evaluations $F_2(\mathbf{x}; \boldsymbol{\xi})$ and $F_2(\mathbf{y}; \boldsymbol{\xi})$ satisfying

$$F_2(\mathbf{x}; 0) = L\|\mathbf{x}\|_1, \qquad F_2(\mathbf{x}; 1) = TL\|\mathbf{x} - \mathbf{u}\|_1 - (T-1)L\|\mathbf{x}\|_1,$$
$$F_2(\mathbf{y}; 0) = L\|\mathbf{y}\|_1, \qquad F_2(\mathbf{x}; 1) = TL\|\mathbf{y} - \mathbf{u}\|_1 - (T-1)L\|\mathbf{y}\|_1.$$

Thus, we have $\mathbb{E}_{\boldsymbol{\xi} \sim \Xi}[F_2(\mathbf{z}; \boldsymbol{\xi})] = f_2(\mathbf{z})$ for all $\mathbf{z} \in \mathbb{R}^d$, which satisfies Assumption 2.7. Moreover, we can verify that both $F_2(\mathbf{z}; 0)$ and $F_2(\mathbf{z}; 1)$ are convex and Lipschitz continuous on $\mathbb{R}$, with the Lipschitz constant $L_2 = 2TL$.

We initialize the algorithm at $\mathbf{x}_0 = \mathbf{1}_d$. The probability of obtaining the identical information from both oracles $F_1$ and $F_2$ with $T$ oracle calls is given by

$$p = \left(1 - \frac{1}{T}\right)^T \geq \frac{1}{e}$$

for all $T \geq 2$. This implies that any SZO algorithm cannot distinguish $f_1(\cdot)$ and $f_2(\cdot)$ in $T$ SZO calls with probability at least $1/e$. Therefore, a near-optimal SZO algorithm $\mathcal{A}$ must achieve the nearly tight function value gaps for both functions with probability $1/e$. In other words, algorithm $\mathcal{A}$ must output $\hat{\mathbf{x}}$ satisfying

$$f_1(\hat{\mathbf{x}}) - f_1^\star \leq \theta_1\left(\frac{\bar{L}}{\underline{L}}, \frac{\bar{s}}{\underline{s}}, T, d\right) \cdot \frac{\sqrt{d}L_1\|\mathbf{x}_0 - \mathbf{x}_{1,*}\|_2}{\sqrt{T}} \quad \text{and} \quad f_2(\hat{\mathbf{x}}) - f_2^\star \leq \theta_2\left(\frac{\bar{L}}{\underline{L}}, \frac{\bar{s}}{\underline{s}}, T, d\right) \cdot \frac{\sqrt{d}L_2\|\mathbf{x}_0 - \mathbf{x}_{2,*}\|_2}{\sqrt{T}},$$

where $\mathbf{x}_{1,\star} \triangleq \arg\min_{\mathbf{x} \in \mathbb{R}^d} f_1(\mathbf{x}) = \mathbf{0}$, $\mathbf{x}_{2,\star} \triangleq \arg\min_{\mathbf{x} \in \mathbb{R}^d} f_2(\mathbf{x}) = \mathbf{u}$, and $\theta_1, \theta_2 : \mathbb{R}^4 \to \mathbb{R}$ are two polylogarithmic functions. Substituting $L_1 = L$ and $L_2 = 2TL$, the corresponding bounds become

$$\|\hat{\mathbf{x}}\|_1 \leq \theta_1\left(\frac{\bar{L}}{\underline{L}}, \frac{\bar{s}}{\underline{s}}, T, d\right) \cdot \frac{d}{\sqrt{T}} \quad \text{and} \quad \|\hat{\mathbf{x}} - \mathbf{u}\|_1 \leq \theta_2\left(\frac{\bar{L}}{\underline{L}}, \frac{\bar{s}}{\underline{s}}, T, d\right) \cdot \sqrt{d}\|\mathbf{1}_d - \mathbf{u}\|_2\sqrt{T}.$$

Since $\mathbf{u} = (1 - 1/T)\mathbf{1}_d$, then the point $\hat{\mathbf{x}}$ must satisfy

$$\|\hat{\mathbf{x}}\|_1 \leq \theta_1\left(\frac{\bar{L}}{\underline{L}}, \frac{\bar{s}}{\underline{s}}, T, d\right) \cdot \frac{d}{\sqrt{T}} \quad \text{and} \quad \|\hat{\mathbf{x}}\|_1 \geq d - \frac{d}{T} - \theta_2\left(\frac{\bar{L}}{\underline{L}}, \frac{\bar{s}}{\underline{s}}, T, d\right) \cdot \frac{d}{\sqrt{T}}.$$

Since the functions $\theta_1$ and $\theta_2$ are poly-logarithmic, there exist a sufficient large $T$ such that

$$\theta_1\left(\frac{\bar{L}}{\underline{L}}, \frac{\bar{s}}{\underline{s}}, T, d\right) \cdot \frac{d}{\sqrt{T}} < d - \frac{d}{T} - \theta_2\left(\frac{\bar{L}}{\underline{L}}, \frac{\bar{s}}{\underline{s}}, T, d\right) \cdot \frac{d}{\sqrt{T}},$$

which leads to contradiction. Hence, we conclude that achieving an ideal parameter-free stochastic zeroth-order algorithm described in the theorem is impossible. $\square$

