# OpenReview forum: "A Parameter-Free and Near-Optimal Zeroth-Order Algorithm for Stochastic Convex Optimization"
_ICML.cc/2025/Conference — ICML 2025 poster_

### Official Review · Reviewer_7pJ3 · 2025-03-02

**Overall Recommendation:** 2

**Summary:**

The paper gives a parameter-free zeroth-order algorithm for convex and Lipschitz continuous stochastic functions $f(x) = E[F(x,\xi)]$ over a convex and compact (section 3/4) or convex and closed (section 5) set. It achieves optimal convergence rates (up to a logarithmic factor) without knowledge of the Lipschitz parameter or the diameter of the constraint set $X$. The parameter-free convergence rates are achieved using an AdaGrad style tuning and the current maximum norm distance to the initial iterate $x_0$. The theoretical findings are supported by illustrating experiments.

**Claims And Evidence:**

Claims are clear and supported.

**Essential References Not Discussed:**

To the best of my knowledge, all relevant literature is discussed.

**Ethical Review Concerns:**

none.

**Experimental Designs Or Analyses:**

The paper contains a small experimental part with numerical experiments illustrating the theoretical findings. To the best of my understanding, the experimental design is sound and valid.

**Methods And Evaluation Criteria:**

Primarily theoretical work, hence, benchmark datasets are not relevant.

**Other Comments Or Suggestions:**

While the paper is very clear and well-written, I think the contribution is rather incremental. Many of the theoretical results share similarities with the results in Ivgi et al and Shamir, up to the point that the proofs follow very similar ideas and structures.

I think the paper would profit a lot from clarifying what its technical contribution and new insights are.

I am happy to revise my evaluation if the authors can convince me that there is a significant difference between existing work and their contribution.

**Other Strengths And Weaknesses:**

Strengths:
- This work is very clearly written.

Weaknesses:
- To the best of my understanding, the results and techniques are rather incremental. Most results follow directly with small modifications (extensions from first- to zeroth-order) of the work by Ivgi et al '23a combined with the zeroth order gradient estimation in Shamir '17. Specifically, the similarities in techniques to the work by Ivgi et al are -to the best of my understanding- very strong.

**Questions For Authors:**

See weaknesses and comments.

**Relation To Broader Scientific Literature:**

The contributions of this paper are rather incremental. (for more details, see weaknesses)

**Theoretical Claims:**

To the best of my understanding, theoretical claims are correct.

---

> ### Author Rebuttal · Authors · 2025-03-29
>
> Thank you for your thoughtful review. We highlight our technical contribution and new insights as follows:
>
> 1. To the best of our knowledge, the proposed POEM is the first parameter-free algorithm for zeroth-order stochastic optimization, which is also mentioned by all other reviewers.
>
> 2. Our method can achieve the near-optimal convergence with the smoothing parameter $\mu_t={\mathcal O}(\sqrt{d/t})$, which is larger than the counterparts in existing works such as $\mu_t={\mathcal O}(\sqrt{d/T})$ (Duchi et al., 2015)  and $\mu_{2t}={\mathcal O}(1/(d^2t^2))$ (Shamir, 2017).
> This is a new observation which improves the numerical stability in the step of finite difference.
>
> 3. For the unbounded domain case, we establish the lower bound (Theorem 5.6) to show that achieving an ideal parameter-free stochastic zeroth-order algorithm is impossible, i.e., the algorithm cannot attain the near-optimal SZO complexity with only logarithmic dependence on problem parameters.
> Our lower bound construction considers the dependence on $d$ in the complexity (construct the $d$ dimensional function), which is more challenging than the lower bound for the first-order method that depends on the 1-dimensional function (Khaled \& Jin, 2024).

---

### Official Review · Reviewer_dAwi · 2025-03-13

**Overall Recommendation:** 4

**Summary:**

This paper introduces a novel parameter-free zeroth-order optimization algorithm named POED for stochastic convex optimization problems. The key idea is to eliminate the need for parameter tuning including learning rate and smoothing parameter. Inspired by difference of gradients (DoG), the proposed method leverages a difference of finite difference strategy to set the learning rate and use an adaptive smoothing parameter. The authors prove that POED achieves near-optimal SZO complexity under bounded domain assumption while impossible to achieve parameter-free algorithm in unbounded setting. Finally, numerical experiments on several datasets demonstrate the superiority of POEM.

## update after rebuttal

Most of my concerns have been addressed during the rebuttal.

**Claims And Evidence:**

The paper claims that (a) POEM is a parameter‐free method, (b) it achieves near‐optimal SZO complexity under convexity, Lipschitz conditions and bounded domain assumptions. To support thest claims, the authors provide rigious theoretical analysis that establish high-probability convergence guarantees. The experimental results also verify the superiority of POEM. Overall, the claims are well supported by both analysis and experiments.

**Essential References Not Discussed:**

N/A

**Experimental Designs Or Analyses:**

The experimental design is straightforward and relevant. The use of standard benchmarks helps validate the algorithm’s practical performance. However, the scope of the experiments is somewhat limited in terms of dataset diversity and scale, as well as the scale of the model scale. More extensive empirical validation could further bolster the claims.

**Methods And Evaluation Criteria:**

POEM is designed to automatically adjust its step size and its smoothing parameter. This design avoids the typical need to manually set these hyperparameters, which is well aligned with problem of parameter-free optimization problems.
The analysis centers on standard metrics in zeroth-order optimization, such as the function value gap and the SZO complexity. These criteria are consistent with the established literature.

**Other Comments Or Suggestions:**

n/a

**Other Strengths And Weaknesses:**

Strenths:

1) To the best of my knowledge, this is the first work to explore a parameter-free zeroth-order optimization.

2)  Comprehensive theoretical analysis with convergence guarantees matching lower bounds.

Weaknesses:

1) The empirical evaluation is limited to a few datasets and could be extended to cover more diverse and large-scale scenarios.

**Questions For Authors:**

1. Can you elaborate on the challenges and potential modifications required to extend POEM to nonconvex optimization problems?

2. What is the computational cost associated with computing the adaptive step size? The computation of the adaptive step size seems need a copy of initial parameters weight. How does this overhead compare with that of standard zeroth-order methods in large-scale applications and do you think there exists a memory-efficient way for the implementation?

3. Recent research has focused on fine-tuning large language models with zeroth-order optimization algorithms while sensitive to the selection of learning rate ([1], [2]). The proposed method seems to be a promising way for solving this problem. Do you think POEM is suitable for large-scale applications like training/fine-tuning large-scale models?

[1] Fine-Tuning Language Models with Just Forward Passes.

[2] Sparse MeZO: Less Parameters for Better Performance in Zeroth-Order LLM Fine-Tuning.

**Relation To Broader Scientific Literature:**

The work is well situated within the current literature and clearly identifies the gap it fills—namely, the lack of parameter-free methods in zeroth-order optimization that achieve near-optimal complexity.

**Theoretical Claims:**

The mian theoretical claims include the convergence rate guarantees for bounded and unbounded domains. The proofs build on well-established lemmas and techniques. I didn't carefully check all the proofs but the theoretical results seem to be sound.

---

> ### Author Rebuttal · Authors · 2025-03-29
>
> **Q1** The empirical evaluation is limited to a few datasets and could be extended to cover more diverse and large-scale scenarios.
>
> **A1**  Thank you for your suggestion.
> We have addressed your comment by including the experiments on the higher-dimensional datasets "qsar" ($d=1,024$, $n=1,687$) from UCI machine learning repository and "gisette" ($d=6,000$, $n=5,000$) from LIBSVM repository.
> Please see link <https://anonymous.4open.science/api/repo/a-5532/file/response-dAwi.pdf?v=78140e95> for the experimental results.
> We can observe that our POEM also performs better than baselines.
>
> **Q2** Can you elaborate on the challenges and potential modifications required to extend POEM to nonconvex optimization problems?
>
> **A2**  Thank you for your question.
> It is still unclear how to extend POEM to nonconvex optimization problem.
> The convexity plays a crucial role in our theoretical analysis:
>
> 1. In Equation (8), we bound the function value gap by using Jensen’s inequality, which relies on the convexity of $f(\cdot)$.
>
> 2. In Equation (9) and the poof of Lemma C.3, we use the first-order condition in Lemma A.1 which requires the convexity of the smooth surrogate $f_\mu(\cdot)$, and the convexity of $f_\mu(\cdot)$ comes from the convexity of the objective $f$ (see Lemma 2.8).
>
> For the nonconvex problem, it is possible to introduce the online-to-nonconvex conversion [1-3] into our framework to attain the nearly-tight upper bound. However, it seems not to be a direct extension. We believe this is a good future direction.
>
> References
>
> [1] Ashok Cutkosky, Harsh Mehta, Francesco Orabona. Optimal stochastic non-smooth non-convex optimization through online-to-non-convex conversion. ICML 2023.
>
> [2] Guy Kornowski, Ohad Shamir. An algorithm with optimal dimension-dependence for zero-order nonsmooth nonconvex stochastic optimization. JMLR 2024.
>
> [3] Kwangjun Ahn, Gagik Magakyan, Ashok Cutkosky. General framework for online-to-nonconvex conversion: schedule-free SGD is also effective for nonconvex optimization. arXiv:2411.07061, 2024
>
> **Q3** What is the computational cost associated with computing the adaptive step size? The computation of the adaptive step size seems need a copy of initial parameters weight. How does this overhead compare with that of standard zeroth-order methods in large-scale applications and do you think there exists a memory-efficient way for the implementation?
>
> **A3** Thanks for your insightful question.
> Each iteration of POEM requires the computational cost of ${\mathcal O}(d)$ to access $||x_t-x_0||$ and $||g_t||$ to determine the step size.
>
> Compared with standard zeroth-order methods, POEM needs to additionally maintain the initial point $x_0$, which requires storing additional $d$ floating point numbers in general.
> However, the domains of many real applications contain the origin.
> In such case, we can simply set $x_0=0$  to avoid the additional memory cost to store $x_0$.
>
> **Q4** Recent research has focused on fine-tuning large language models with zeroth-order optimization algorithms while sensitive to the selection of learning rate ([1], [2]). The proposed method seems to be a promising way for solving this problem. Do you think POEM is suitable for large-scale applications like training/fine-tuning large-scale models?
>
> **A4** Thanks for your constructive suggestions.
> This work focuses on convex optimization, while training/fine-tuning large-scale models typically corresponds to nonconvex optimization.
> As we mentioned in A2, our theory cannot be directly extended to the nonconvex case.
> Following your suggestion, we are currently attempting to adapt POEM to the fine-tuning of large-scale models. We have not yet determined whether POEM is suitable for this task.

---

### Official Review · Reviewer_7JXP · 2025-03-16

**Overall Recommendation:** 4

**Summary:**

This paper proposes a parameter-free stochastic zeroth-order method that achieves near-optimal rate in the convex setting with a bounded domain. The authors also consider the unbounded domain case and prove that it is impossible to construct an ideal parameter-free algorithm in this setting.

**Claims And Evidence:**

Yes

**Essential References Not Discussed:**

The literature is sufficiently covered.

**Experimental Designs Or Analyses:**

The experimental design is reasonable.

**Methods And Evaluation Criteria:**

Yes

**Other Comments Or Suggestions:**

Figure 3 has not been mentioned in the main text. Add a description to it.

**Other Strengths And Weaknesses:**

Strengths:
- Clear writing with a nice flow of ideas.
- The first parameter-free stochastic zeroth-order method in the convex setting, which is a solid contribution
- The impossibility result is interesting which reveals the fundamental necessity of a bounded domain.
- The robustness of the parameter setting of POEM is numerically verified

Weaknesses:
NA

**Questions For Authors:**

NA

**Relation To Broader Scientific Literature:**

This paper considers parameter-free stochastic optimization, several related works are published in past ICML events.

**Theoretical Claims:**

I do not have the time to check all the proofs. To what I have verified, everything looks fine.

---

> ### Author Rebuttal · Authors · 2025-03-29
>
> Thank you for your positive feedback and appreciation of our work.
>
> **Q1** Figure 3 has not been mentioned in the main text. Add a description to it.
>
> **A1** We sincerely thank you for your careful review.
> We will modify the text in lines 436-438 of the left column to include a clear reference to Figure 3.

---

### Official Review · Reviewer_ZPYR · 2025-03-17

**Overall Recommendation:** 3

**Summary:**

This paper addresses the stochastic optimization problem
$$
\min_{x \in \mathcal{X}} \mathbb{E}_{\xi}[F(x, \xi)],
$$
where each function $F(\cdot, \xi)$ is convex and $L$-Lipschitz, and
$\mathcal{X}$ is a simple convex set in $\mathbb{R}^d$. The authors propose a
parameter-free zeroth-order optimization method that achieves
$\epsilon$-accuracy (in terms of the objective function) with high probability
in $\tilde{O}(\frac{d L^2 D^2}{\epsilon^2})$ stochastic function evaluations&mdash;nearly
optimal up to logarithmic factors.

The proposed method builds on DoG (Ivgi et al., 2023), a parameter-free
stochastic gradient method with distance adaptation, applied to a randomized
smoothing of the objective.

The authors also present a version of their algorithm for unbounded domains,
assuming a good estimate of $L$ is available. Additionally, they
establish a lower bound demonstrating that a fully parameter-free algorithm with
complexity scaling as $\tilde{O}(\frac{d L^2 D_0^2}{\epsilon^2})$ is impossible,
implying that any efficient parameter-free method must depend on the domain
diameter $D$, rather than just the initial distance $D_0$ to the solution.

**Claims And Evidence:**

The claims made in the paper are well-supported, with each theorem and lemma
accompanied by a corresponding proof.

**Essential References Not Discussed:**

I have not identified any essential references missing from the discussion.

**Experimental Designs Or Analyses:**

I examined the experiments and found no major issues.

**Methods And Evaluation Criteria:**

The proposed methods and evaluation criteria are appropriate for the problem at
hand.

**Other Comments Or Suggestions:**

1. The claims that the method is "parameter-free" in the Abstract and early
   sections of the paper require clarification. The algorithm is truly
   parameter-free only when the domain is bounded and the complexity bound
   depends on the domain's diameter. For unbounded problems (or when results
   depend on the initial distance $D_0$ rather than $D$), the method requires an
   upper bound on the stochastic gradient norm.

1. There is a mistake in the formula for $\mu_t$ in Algorithm 1, which makes the
   final bound in Proposition 4.7 not scale-invariant. The correct formula is
   likely $\mu_t = \bar{r}_t \sqrt{\frac{d}{t + 1}}$.

1. Assumption 2.6 is missing "for all $x, y \in \mathbb{R}^d$".

1. Line 163: The claim that "this is more challenging" is somewhat misleading.
   In principle, $\mu_t$ can be arbitrarily small, even zero, in which case
   finite differences reduce to a directional derivative. While choosing a very
   small $\mu_t$ may degrade numerical stability in practice, this is a separate
   issue unrelated to the theoretical complexity bounds.

1. Lemma A.4: The reference to Shamir (2017, Lemma 9) seems incorrect.

1. Typos: "differed" (line 169), "soothing" (line 267).

**Other Strengths And Weaknesses:**

**Strengths:**

1. This is the first (to my knowledge) parameter-free algorithm for zeroth-order
   stochastic optimization. The presentation is concise and generally clear.

1. The lower bound in Theorem 5.6 is a valuable theoretical contribution (though
   I have not fully verified its correctness).

**Weaknesses:**

1. The paper primarily combines existing results from (Ivgi et al., 2023) and
   (Shamir, 2017). The proposed algorithm is essentially DoG applied to a
   smoothed objective with a carefully chosen smoothing parameter $\mu$. The
   complexity bound follows directly from prior DoG results (Ivgi et al., 2023)
   and  established bounds on the second moment of stochastic gradients for
   the smooth approximation (Shamir, 2017). The main novelty&mdash;adapting
   $\mu_t$ at each iteration rather than fixing it in advance&mdash;introduces
   only minor modifications to the DoG analysis. While this is a useful
   refinement, it does not introduce a fundamentally new idea.

**Questions For Authors:**

1. The paper assumes that the original objective is Lipschitz. What if it is
   Lipschitz-smooth? Can the proposed method still be applied, and what would
   its convergence rate be?

**Relation To Broader Scientific Literature:**

This paper integrates two established optimization techniques:

1. *Randomized smoothing*, which approximates the original objective via a
   smooth surrogate, enabling efficient stochastic gradient estimation (Duchi et
   al., 2012; Yousefian et al., 2012; Shamir, 2017; Nesterov & Spokoiny, 2017;
   Gasnikov et al., 2022; Lin et al., 2022). The specific smoothing method used
   here&mdash;uniform smoothing over the Euclidean ball&mdash;has been
   extensively studied in (Duchi et al., 2012; Shamir, 2017; Gasnikov et al.,
   2022).

2. The *DoG algorithm* (Ivgi et al., 2023), which enables efficient
   parameter-free stochastic optimization of convex functions.

Much of the analysis in this paper builds on the proofs in (Ivgi et al., 2023),
leveraging known properties of randomized smoothing from (Shamir, 2017).

**Theoretical Claims:**

I reviewed the main theoretical claims and assessed the general proof techniques
but did not verify all details rigorously.

---

> ### Author Rebuttal · Authors · 2025-03-29
>
> **Q1** The main novelty—adapting at each iteration rather than fixing it in advance—introduces only minor modifications to the DoG analysis. While this is a useful refinement, it does not introduce a fundamentally new idea.
>
> **A1** Thank you for your thoughtful review. We highlight our novelty as follows:
>
> 1. Our method can achieve the near-optimal convergence with the smoothing parameter $\mu_t=\mathcal{O}(\sqrt{d/t})$, which is larger than the counterparts in existing works such as $\mu\_t=\mathcal{O}(\sqrt{d/T})$(Shamir, 2017)  and $\mu\_{2t}=\mathcal{O}(1/(d^2t^2))$ (Duchi et al., 2015). This is a new observation which improves the numerical stability.
>
> 2. For the unbounded domain case, we establish the lower bound (Theorem 5.6) to show that achieving an ideal parameter-free stochastic zeroth-order algorithm is impossible, i.e., the algorithm cannot attain the near-optimal SZO complexity with only logarithmic dependence on problem parameters. Our lower bound construction considers the dependence on $d$ in the complexity (construct the $d$ dimensional function), which is more challenging than the lower bound for the first-order method that depends on the 1-dimensional function (Khaled & Jin, 2024).
>
> **Q2** The algorithm is truly parameter-free only when the domain is bounded and the complexity bound depends on the domain's diameter.
>
> **A2** Thank you for your suggestion. We will clarify the claim of "parameter-free" in revision.  For the unbounded problems, we provide the lower bound to show that it is impossible to achieve the near-optimal and parameter-free algorithm. Please see the second point in the last response.
>
> **Q3** There is a mistake in the formula $\mu_t$ for Algorithm 1, which makes the final bound in Proposition 4.7 not scale-invariant. The correct formula is likely $\mu\_t=\bar{r}\_t\sqrt{\frac{d}{t+1}}$.
>
> **A4** Thank you for your careful review. You're right that the smoothing parameter should be $\mu_t = \bar{r}\_t \sqrt{\frac{d}{t+1}}$.  The result of Lemma 4.5 should be modified to
> $$
> \sum_{k=0}^{t-1}2L \bar{r}\_k \mu\_k\leq 4L \bar{r}\_{t-1}^2\sqrt{dt},
> $$
> where the exponent of $\bar{r}\_{t-1}$ changes from $1$ to $2$. Thus, Proposition 4.7 should be stated as:
> $$
> f(\bar{x}\_t)-f(x^*)\leq\frac{16\theta\_{t,\delta}(\bar{r}\_t+s\_0)\big(\sqrt{G\_{t-1}}+Ld+L\sqrt{dt}\big)}{\sum_{k=0}^{t-1} \bar{r}\_k/ \bar{r}\_t},
> $$
> which is scale-invariant.
> Based on above modification, the main result (Theorem 4.9) still holds.
> We also update our implementation and empirical results are very similar to previous ones. Please see the link <https://anonymous.4open.science/api/repo/a-5532/file/response-ZPYR.pdf?v=abeb97c6>.
>
> **Q4** Assumption 2.6 is missing "for all $ x,y\in\mathbb{R}^d$".
>
> **A4** Thank you for your careful review. We will involve the description for these variables in revision.
>
> **Q5** Line 163: The claim that "this is more challenging" is somewhat misleading.  In principle, $\mu_t$ can be arbitrarily small, even zero, in which case finite differences reduce to a directional derivative. While choosing a very small may degrade numerical stability in practice, this is a separate issue unrelated to theoretical complexity bounds.
>
> **A5** Thank you for your valuable suggestion. In revision, we will clarify that the choice of $\mu_t$ is important to the numerical stability, rather than the theoretical complexity bounds.
>
> **Q6** Lemma A.4: The reference to Shamir (2017, Lemma 9) seems incorrect.
>
> **A6** Thank you for your careful review. The result of Lemma A.4 appears at the beginning of the proof of Shamir (2017, Lemma 9), which can be found at the bottom of page 6 (not the final result of Lemma 9). We will clarify this point in revision.
>
> **Q7** Typos: "differed" (line 169),  "soothing" (line 267).
>
> **A7** We appreciate your careful review. We will correct the typos in revision.
>
> **Q8** The paper assumes that the original objective is Lipschitz. What if it is Lipschitz-smooth? Can the proposed method still be applied, and what would its convergence rate be?
>
> **A8** Thank you for your insightful question.
>
> For the bounded setting, our proposed method remains applicable. We assume that $F(\cdot;\xi)$ is $M$-smooth, i.e., $\Vert \nabla F(x)-\nabla F(y)\Vert \leq M\Vert x-y\Vert$ for all $x,y \in \mathcal{X}$.
> Let $x^*\in \mathcal{X}$ be the solution.
> It follows that $\Vert \nabla F(x)\Vert\leq\Vert\nabla F(x^*)\Vert+\Vert\nabla F(x)-\nabla F(x^*)\Vert\leq\Vert\nabla F(x^*)\Vert+M\Vert x-x^*\Vert\leq\Vert\nabla F(x^*)\Vert+MD_\mathcal{X}$ for all $x\in \mathcal{X}$.
> Thus, the function $F(\cdot;\xi)$ is $(\Vert\nabla F(x^*)\Vert+M D_\mathcal{X})$-Lipchitz on the domain $\mathcal{X}$.
> Hence, we can directly apply Theorem 4.9 to achieve the convergence rate.
>
> For the unbounded setting, the additional assumption such as bounded variance for the stochastic gradient (Ghadimi & Lan, 2013) is typically required.
> It seems that our analysis cannot be directly applied to this case.

---

### Decision · Program_Chairs · 2025-05-01

**Decision:**

Accept (poster)

**Comment:**

This paper proposes a parameter-free zeroth-order algorithm for stochastic optimization. It is an extension of some recent parameter-free first-order algorithms. It addresses the important issue that under certain scenarios first-order oracle is not available but only zeroth-order oracle is available. This is a nice contribution to the literature of parameter-free algorithms for stochastic optimization.